# Simulation-based Bayesian Inference from Privacy Protected Data

**Yifei Xiong**                                                          *xiong173@purdue.edu*
*Department of Statistics*
*Purdue University*

**Nianqiao Phyllis Ju**                                                  *nianqiao@purdue.edu*
*Department of Statistics*
*Purdue University*

**Sanguo Zhang**                                                         *sgzhang@ucas.ac.cn*
*School of Mathematical Sciences*
*University of Chinese Academy of Sciences*

**Reviewed on OpenReview:** *https://openreview.net/forum?id=SB7JzhDG45*

## Abstract

Many modern statistical analysis and machine learning applications require training models on sensitive user data. Under a formal definition of privacy protection, differentially private algorithms inject calibrated noise into the confidential data or during the data analysis process to produce privacy-protected datasets or queries. However, restricting access to only privatized data during statistical analysis makes it computationally challenging to make valid statistical inferences. In this work, we propose simulation-based inference methods from privacy-protected datasets. In addition to sequential Monte Carlo approximate Bayesian computation, we adopt neural conditional density estimators as a flexible family of distributions to approximate the posterior distribution of model parameters given the observed private query results. We illustrate our methods on discrete time-series data under an infectious disease model and with ordinary linear regression models. Illustrating the privacy-utility trade-off, our experiments and analysis demonstrate the necessity and feasibility of designing valid statistical inference procedures to correct for biases introduced by the privacy-protection mechanisms.

## 1 Introduction

**Motivation.** Many AI systems require collecting and training on massive amounts of personal information (such as income, disease status, location, purchase history, etc.). Despite unprecedented data collection efforts by companies, governments, researchers, and other agencies, oftentimes, data collectors have to lock the data inside their own databases due to privacy concerns. Differential privacy (DP) provides a mathematical definition for the protection of individual data (Drechsler et al., 2024). Under this framework, privacy-protecting procedures (i.e., DP algorithms) have enabled data collectors, such as tech companies, the US Census Bureau, and social scientists, to share research data in a wide variety of settings while protecting the privacy of individual users. Privacy researchers typically collect confidential data and then inject calibrated random noise into the confidential data to achieve the desired levels of privacy protection. Some algorithms aim for DP data analysis, resulting in DP optimizations (Arora et al., 2023; Bassily et al., 2021), approximations (Chaudhuri et al., 2013; Bie et al., 2023), or predictions (Rho et al., 2023); while other algorithms produce DP datasets (or descriptive statistics), which enables data sharing across research teams and entities. Examples of the latter include the 2020 US Census (Abowd et al., 2022; Gong et al., 2022;

Drechsler, 2023), the Facebook URL dataset (Evans & King, 2023), and New York Airbnb Open Data (Guo & Hu, 2023). Our work tackles this data-sharing regime: we aim to make valid statistical inferences with privatized data, and we place a special focus on using complex models such as continuous-time Markov jump processes.

Although the DP data-sharing regime corrupts confidential information in order to satisfy privacy, since the probabilistic design of such mechanisms can be publicly known, in principle analysts can still conduct reliable estimation and uncertainty quantification by accounting for bias and noises introduced during privatization. However, in practice, valid inference based on privatized data is a challenge that requires the revision of existing statistical methods designed originally for confidential data (Foulds et al., 2016). Even for well-understood procedures such as ordinary linear regression and generalized linear models, adding the extra layer of privacy protection has introduced new theoretical and methodological questions in statistics (Cai et al., 2021; Alabi & Vadhan, 2022; Li et al., 2023; Barrientos et al., 2019).

However, for complex models, even the confidential data likelihood functions are intractable or time-consuming to compute. Accounting for privacy noise on top of that is a formidable challenge. Without developing valid inference procedures under this regime, we must either make biased estimations or restrict ourselves to simple models. In this work, we propose methods to estimate the parameters of complex models that underlie privacy-protected data.

**Related works.** There is a fast-growing literature on statistical inference under differential privacy. Early work by Williams & McSherry (2010) first proposed using Bayesian inference to handle DP noise. Since then, various Bayesian methods have been developed for privatized data. Markov Chain Monte Carlo (MCMC) methods have been proposed in some specific models and priors, such as exponential family distributions (Bernstein & Sheldon, 2018), Bayesian linear regression (Bernstein & Sheldon, 2019) and generalized linear models (Kulkarni et al., 2021). As for generic algorithms, Ju et al. (2022) proposed a data-augmentation MCMC strategy to overcome the intractable marginal likelihood resulting from privatization, and Gong (2022) derived point estimates of the posterior distribution using the Expectation-Maximization algorithm. Several frequentist inference methods have also been developed. Karwa et al. (2015) employed a parametric bootstrap method to construct confidence intervals for the model parameters of the log-linear model. Awan & Wang (2023) proposes simulation-based inference methods for hypothesis testing and confidence intervals, for frequentist inference.

To the best of our knowledge, only three papers (Waites & Cummings, 2021; Lee et al., 2022; Su et al., 2023) have incorporated normalizing flows (Kobyzev et al., 2020; Papamakarios et al., 2021) and DP, and both works design DP-versions of normalizing flow. In contrast, our work uses flow-based methods as a neural density estimation tool to analyze DP-protected data.

**Our contributions.** In this work, we propose several likelihood-free inference methods that make statistical inferences from privacy-protected data. First, we highlight that sequential Monte Carlo approximate Bayesian computation (SMC-ABC) can be used for this purpose, which improves the current practice of using ABC (Gong, 2022). Next, we propose sequential private posterior estimation (SPPE) and sequential private likelihood estimation (SPLE), two sequential neural density estimation methods to approximate the private data posterior distribution with normalizing flows. Unlike likelihood-based methods to learn from private data, SMC-ABC, SPPE, and SPLE require only simulations from a generative model for confidential data, and hence are applicable to more models. We demonstrate the efficiency of our methods on an infectious disease model using synthetic and several real disease outbreak data. SPPE and SPLE require fewer numbers of simulations from the data model to achieve the same inference results compared with SMC-ABC. We also propose a privacy mechanism for the release of the infection curve. Our experiments also demonstrate the privacy-utility trade-off in linear regression, where we compared the proposed likelihood-free methods with data augmentation MCMC, a likelihood-based inference method. The code is available on GitHub[1].

---

[1] https://github.com/Yifei-Xiong/Simulation-based-Bayesian-Inference-from-Privacy-Protected-Data

## 2 Background and challenges

Let $\theta \in \Theta$ be the model parameter and $x = (x_1, \cdots, x_n) \in \mathbb{X}^n$ represent the confidential database, where $n$ is the sample size. We model the database with some likelihood function $f(x \mid \theta)$. This confidential data model can be a 'simulator' whose likelihood can be impossible to compute. In the confidential data setting, our interest is to approximate the posterior distribution $\pi(\theta \mid x^o) \propto \pi(\theta) f(x^o \mid \theta)$ where $x^o$ is the observed dataset and $\pi(\theta)$ is the prior. In this section, we first review some key approaches for likelihood-free inference in the confidential data setting and then describe output perturbation mechanisms to generate differentially private queries $s_{\mathrm{dp}}$. Finally, we explain why learning $\theta$ given observed $s_{\mathrm{dp}}^o$ is challenging. We will present new methods for likelihood-free inference in the privatized data setting in Section 3.

### 2.1 Likelihood-free inference for confidential data

Likelihood-free inference methods require only the ability to generate data from a simulator model, instead of evaluating its likelihood.

**Approximate Bayesian Computation (ABC) and sequential Monte Carlo ABC (SMC-ABC).** ABC-based methods circumvent likelihood evaluations with simulations. In its most basic form, a set of parameters are sampled from the prior $\theta^{(i)} \sim \pi(\theta)$ and a data set $x^{(i)}$ is simulated from the model for each $\theta^{(i)}$. The samples $\theta^{(i)}$ such that $x^{(i)}$ is close to the observed data $x^o$ provide an approximation to the posterior $\pi(\theta \mid x^o)$. It is also common to choose some summary statistics $s$ and to retain samples according to distance between each $s(x^{(i)})$ and $s(x^o)$. The efficiency of ABC is controlled by the closeness between the prior distribution, which can be viewed as its proposal distribution, and the posterior distribution, which is the target distribution. SMC-ABC (Sisson et al., 2007; Beaumont et al., 2009) improves ABC by specifying a sequence of intermediate target distributions between the prior and the posterior, and thus the $\theta^{(i)}$ samples can gradually evolve towards the target distribution. The sequential updates in SMC-ABC propagate and reweight the parameter samples with importance sampling.

**Neural density estimation.** In contrast to sampling-based approximations like ABC and SMC-ABC, we can pursue density approximation of $\pi(\theta \mid x^o)$ with neural networks. We illustrate a neural estimator for the posterior density $\pi(\theta \mid x)$. Let $\{q_\phi(\theta \mid x)\}_\phi$ denote the collection of densities representable by an architecture $q$ and weight parameters $\phi$. We can train the neural estimator to minimize an ideal loss function

$$\hat{\phi} = \arg\min_\phi \mathbb{E}_{p(\theta, x)} \left[ -\log q_\phi(\theta \mid x) \right] = \arg\min_\phi \int_{\Theta \times \mathbb{X}^n} \left[ -\log q_\phi(\theta \mid x) \right] \cdot \pi(\theta) f(x \mid \theta) \mathrm{d}\theta \mathrm{d}x.$$

This loss is an expected value of $-\log q_\phi$ under the joint distribution $p(\theta, x) = \pi(\theta) f(x \mid \theta)$. When $f(x \mid \theta)$ is intractable, the expectation is also intractable, and training relies on Monte Carlo approximations of the integral using samples $\{(\theta^{(j)}, x^{(j)})\}_{i=1}^N$. In sequential neural density estimation, we can sequentially improve the Monte Carlo estimation of the loss function (Papamakarios & Murray, 2016; Lueckmann et al., 2017; Greenberg et al., 2019). Later, in Section 3.4, we will use randomized quasi-Monte Carlo (RQMC) methods (Owen, 1997a;b) as a drop-in replacement for standard Monte Carlo to reduce variance. Other neural density approaches include approximating likelihood functions (Papamakarios et al., 2019) or likelihood ratios (Miller et al., 2022). We refer the readers to (Cranmer et al., 2020; Lueckmann et al., 2021) for systematic reviews of neural density approximations.

A notable neural density estimation method is normalizing flows (NF) (Dinh et al., 2017; Papamakarios et al., 2017; Durkan et al., 2019; Papamakarios et al., 2021), which are adopted in our experiments. NF start with a simple base density (e.g. multivariate Normal) and push it through a sequence of invertible and differentiable transformations. The resulting distribution is richer and more complex. We choose NF because of their expressive power and popularity in probabilistic modeling.

### 2.2 Differential privacy

Given confidential data $x$, let $\eta$ be a randomized algorithm to produce a 'differentially private statistic' $s_{\mathrm{dp}}$ from $x$. We also use $\eta(s_{\mathrm{dp}} \mid x)$ to denote the conditional density of the private output $s_{\mathrm{dp}}$ given the

confidential data $x$. Intuitively, a procedure is private when perturbing one individual's response in the dataset leads to only a small change in the algorithm's outcome. This is characterized by the $\epsilon$-DP definition from Dwork et al. (2006), based on neighboring databases.

**Definition 1** ($\epsilon$-DP)**.** We say $x, x' \in \mathbb{X}^n$ are 'neighboring' databases if they differ by one and only one data record. Denote this by $d(x, x') \leq 1$. A privacy mechanism with conditional density $\eta$ satisfies $\epsilon$-DP if, for all possible values of $s_{\mathrm{dp}}$ and for neighboring datasets, the following probability ratio is bounded:

$$\frac{\int_A \eta(s_{\mathrm{dp}} \mid x) \, \mathrm{d}s_{\mathrm{dp}}}{\int_A \eta(s_{\mathrm{dp}} \mid x') \, \mathrm{d}s_{\mathrm{dp}}} \leq \exp(\epsilon), \quad \forall A \subseteq \mathrm{Range}(\eta). \tag{1}$$

The parameter $\epsilon$ is referred to as the *privacy loss budget*, and it plays a pivotal role in determining the extent to which $s_{\mathrm{dp}}$ discloses information about $x$: Larger values of $\epsilon$ correspond to reduced privacy guarantees, whereas $\epsilon = 0$ signifies perfect privacy.

Output perturbation methods achieve privacy by first computing a query $s : \mathbb{X}^n \to \mathbb{S}$ (such as mean, median, histogram, contingency tables) of the database and then releasing $s(x)$ with added noise. To satisfy $\epsilon$-DP, the query $s$ must have finite sensitivity.

**Definition 2** (Global sensitivity (Nissim et al., 2007))**.** The $L_p$ sensitivity of a function $s$, denoted $\Delta_p(s)$, is the maximum $L_p$-norm change in the function's value between neighboring databases $x$ and $x'$, namely

$$\Delta_p(s) = \max_{d(x,x')=1} \|s(x) - s(x')\|_p .$$

A common output perturbation technique is the Laplace mechanism.

**Proposition 3** (Laplace mechanism (Dwork et al., 2006))**.** *For a real-valued query* $s : \mathbb{X}^n \to \mathbb{S}$*, adding zero-centered Laplace noise with parameter* $\Delta_1(s)/\epsilon$ *achieves* $\epsilon$*-DP.*

Chaudhuri et al. (2013) provides a multivariate version of the Laplace mechanism. Other mechanisms include the exponential mechanism and the Gaussian mechanism (Liu, 2018). There are also relaxations of $\epsilon$-DP, such as $(\epsilon, \delta)$-DP and Gaussian DP (Dong et al., 2022).

We exploit output perturbation on summary statistics in two ways. From an inference perspective, when $s(x)$ is a sufficient statistic for $\theta$, the confidential data posterior $\pi(\theta \mid x)$ is fully characterized by $s(x)$ and the prior. This is the rationale for choosing $\pi(\theta \mid s_{\mathrm{dp}})$ as the target distribution and for the posterior from perturbed data to be still informative about the model parameters. From a computation perspective, since many summary statistics are of lower dimension than the confidential data, this can facilitate efficient computations. In Section 3.4, we leverage the low dimensionality of $s_{\mathrm{dp}}$ by incorporating quasi-Monte Carlo techniques.

## 2.3 Bayesian inference with privatized queries

In the private data setting, our goal is to approximate the posterior distribution of $\theta$ given privatized data $s_{\mathrm{dp}}$. A key challenge with data analysis on privatized data via $\pi(\theta \mid s_{\mathrm{dp}})$ is the intractable private data marginal likelihood.

$$f(s_{\mathrm{dp}} \mid \theta) = \int_{\mathbb{X}^n} f(x \mid \theta)\eta(s_{\mathrm{dp}} \mid x)\mathrm{d}x. \tag{2}$$

When the confidential data likelihood $f(x \mid \theta)$ can be evaluated, Ju et al. (2022) proposed a data-augmentation MCMC algorithm to approximate the doubly-intractable private data posterior

$$\pi(\theta \mid s_{\mathrm{dp}}) \propto f(s_{\mathrm{dp}} \mid \theta)\pi(\theta). \tag{3}$$

The data augmentation strategy circumvents the intractability of evaluating Equation (2) by working with the joint posterior distribution $p(\theta, x \mid s_{\mathrm{dp}}) \propto f(x \mid \theta)\pi(\theta)\eta(s_{\mathrm{dp}} \mid x)$ instead of the marginal posterior in Equation (3).

In this work, we study the challenging scenarios when the confidential likelihood $f(x \mid \theta)$ is intractable. This situation is 'triply intractable', as there are three levels of intractability in $f(x \mid \theta)$, $f(s_{\mathrm{dp}} \mid \theta)$, and $\pi(\theta \mid s_{\mathrm{dp}})$. Data augmentation MCMC, which includes an inference step to sample from $\pi(\theta \mid x)$, is no longer applicable in the 'triply intractable' setting.

# 3 Proposed methods

We present methods for likelihood-free inference for the triply intractable posterior distribution $\pi(\theta \mid s_{\mathrm{dp}})$. First and foremost, we recognize that SMC-ABC methods are viable solutions to approximate the private data posterior, as long as the privacy mechanism is publicly known and can be replicated by the data analyst.

Next, under the same assumptions, we focus on two neural density approximation methods that leverage state-of-the-art simulation-based inference methods and can be more efficient than the SMC-ABC baseline. We present two complementary approaches: a) Section 3.2: approximate the private data marginal likelihood $f(s_{\mathrm{dp}} \mid \theta)$ in Equation (2), and b) Section 3.3: approximate $\pi(\theta \mid s_{\mathrm{dp}})$ from Equation (3) directly.

## 3.1 Adapting SMC-ABC to the private data setting

A data analyst can create a (hierarchical) private data simulator for $s_{\mathrm{dp}}$ by combining the confidential data simulator $x \sim f(\cdot \mid \theta)$ and a tractable output perturbation mechanism (e.g. the Laplace mechanism) with $s_{\mathrm{dp}} \sim \eta(\cdot \mid x)$. With this simulator for $s_{\mathrm{dp}}$, a rejection ABC algorithm can simulate parameter samples $\{\theta^{(i)}\}_{i=1,\dots,N}$ from the prior, and then keep samples correspond to a simulated private query $s_{\mathrm{dp}}^{(i)}$ close to the observed value $s_{\mathrm{dp}}^o$. This insight has been exploited in Gong (2022), using ABC to approximate $\pi(\theta \mid s_{\mathrm{dp}}^o)$. We point out that SMC-ABC can also be used for this purpose. We include a description of the algorithm in Appendix C for completeness.

Both SMC-ABC and ABC aim to target the exact posterior distribution, unlike neural density estimation methods which insist that the approximation comes from some parametric family $q_\phi$. For this reason, we will use SMC-ABC as a baseline to test and validate our methods in Section 4.

## 3.2 Private data likelihood estimation

Our first neural density approximation strategy is to approximate the private data marginal likelihood $f(s_{\mathrm{dp}} \mid \theta)$ with a neural network denoted by $q_\phi(s_{\mathrm{dp}} \mid \theta)$. When training $q_\phi(s_{\mathrm{dp}} \mid \theta) \approx f(s_{\mathrm{dp}} \mid \theta)$, we aim to minimize their average KL divergence under the prior $\pi(\theta)$, corresponding to minimizing

$$\mathbb{E}_{\pi(\theta)}\left[\mathcal{D}_{\mathrm{KL}}\left(f(s_{\mathrm{dp}} \mid \theta) \| q_\phi(s_{\mathrm{dp}} \mid \theta)\right)\right]. \tag{4}$$

After some derivations (in Appendix A), Equation (4) is equivalent to $\mathbb{E}_{p(\theta, s_{\mathrm{dp}})}\left[-\log q_\phi(s_{\mathrm{dp}} \mid \theta)\right]$ up to a constant independent of $\phi$. The expectation is taken with respect to the prior and the private data generating process $p(\theta, s_{\mathrm{dp}}) = \pi(\theta) \cdot f(s_{\mathrm{dp}} \mid \theta)$, which is intractble. To facilitate computations, let's write Equation (4) with respect to the confidential data generating process, resulting in

$$\ell_{\mathrm{PLE}}(\phi) = \mathbb{E}_{p(\theta, x)}\left[-\int_{\mathbb{S}} \eta(s_{\mathrm{dp}} \mid x) \log q_\phi(s_{\mathrm{dp}} \mid \theta) \mathrm{d}s_{\mathrm{dp}}\right]. \tag{5}$$

With $\widehat{\phi} = \arg\min \ell_{\mathrm{PLE}}(\phi)$, the resulting private data likelihood estimation is $q_{\widehat{\phi}}(s_{\mathrm{dp}} \mid \theta)$. Since Equation (5) is an expectation with respect to a tractable distribution, we can approximate the integral with Monte Carlo methods, discussed in Section 3.4.

When the primary inference goal is the maximum likelihood estimator, one can approximate it with $\widehat{\theta}_{\mathrm{MLE}} = \arg\max_\theta q_{\widehat{\phi}}(s_{\mathrm{dp}} \mid \theta)$. Under the Bayesian paradigm, the posterior approximation of Equation (3) can be $\widehat{\pi}_{\mathrm{PLE}}(\theta) \propto \pi(\theta) q_{\widehat{\phi}}(s_{\mathrm{dp}} \mid \theta)$. Quantities such as posterior median, mean, and credible regions can be estimated accordingly.

## 3.3 Private data posterior estimation

Now we approximate the private data posterior $\pi(\theta \mid s_{\mathrm{dp}})$ in Equation (3) directly with a neural posterior estimator $q_\phi(\theta \mid s_{\mathrm{dp}})$, bypassing the synthetic likelihood step in Section 3.2. To find $q_\phi(\theta \mid s_{\mathrm{dp}}) \approx \pi(\theta \mid s_{\mathrm{dp}})$, let us minimize their KL divergence

$$\mathcal{D}_{\mathrm{KL}}\left(\pi(\theta \mid s_{\mathrm{dp}}) \| q_\phi(\theta \mid s_{\mathrm{dp}})\right) = \mathbb{E}_{\pi(\theta \mid s_{\mathrm{dp}})}\left[\log \frac{\pi(\theta \mid s_{\mathrm{dp}})}{q_\phi(\theta \mid s_{\mathrm{dp}})}\right].$$

As shown in Section A, this is equivalent to

$$\ell_{\text{PPE}}(\phi) = \mathbb{E}_{p(\theta,x)}\left[-\int_{\mathbb{S}} \eta(s_{\text{dp}} \mid x) \log q_\phi(\theta \mid s_{\text{dp}}) \mathrm{d}s_{\text{dp}}\right]. \tag{6}$$

Many Bayesian problems use uninformative priors $\pi(\theta)$, which are dispersed in the parameter space. As a result, when employing a naive Monte Carlo strategy that uses the prior as the proposal to approximate expectations in Equation (5) or Equation (6) has most of its samples falling in low-density regions, leading to inefficient estimation due to high variance. To address this issue, importance sampling utilizes a proposal distribution $\tilde{p}$ to change the bases of integration and thus reduce the variance of numerical integration. The automatic posterior transformation (APT) method (Greenberg et al., 2019) uses $\tilde{p}(\theta, x) = \tilde{p}(\theta) f(x \mid \theta)$ as proposal and has (unnormalized) importance weights

$$\tilde{q}_\phi(\theta \mid s_{\text{dp}}) \propto q_\phi(\theta \mid s_{\text{dp}}) \frac{\tilde{p}(\theta)}{\pi(\theta)}. \tag{7}$$

Based on this idea, we design a modified loss of Equation (6) that leverages the APT framework to improve efficiency in posterior estimation. Specifically, we propose the following loss function

$$\ell_{\text{PPE-A}}(\phi) = \mathbb{E}_{\tilde{p}(\theta,x)}\left[-\int_{\mathbb{S}} \eta(s_{\text{dp}} \mid x) \log \tilde{q}_\phi(\theta \mid s_{\text{dp}}) \mathrm{d}s_{\text{dp}}\right], \tag{8}$$

and it is useful for the sequential training strategy we introduce in Section 3.5. The derivation of Equation (8) is in Appendix A.

### 3.4 Nested RQMC estimators

We can inspect the general form of loss functions in Equations (5) and (8) with

$$\ell(\phi) = -\mathbb{E}_{\tilde{p}(\theta,x)}\left[\int_{\mathbb{S}} \eta(s_{\text{dp}} \mid x) g(s_{\text{dp}}, \theta) \mathrm{d}s_{\text{dp}}\right]. \tag{9}$$

Here $g(s_{\text{dp}}, \theta) = \log \tilde{q}_\phi(\theta \mid s_{\text{dp}})$ for PPE-A and $g(s_{\text{dp}}, \theta) = \log q_\phi(s_{\text{dp}} \mid \theta)$ for PLE. Equation 9 is a double integral, where the outer expectation is with respect to some proposal distribution $\tilde{p}$ and the inner integral involves the privacy mechanism $\eta$ and the neural estimator $q_\phi$.

With independent and identically distributed (i.i.d.) samples from the joint density $\{(\theta^{(i)}, x^{(i)})\}_{i=1}^N \sim \tilde{p}(\theta, x) = \tilde{p}(\theta) f(x \mid \theta)$, we can unbiasedly approximate Equation (9) with

$$\widehat{\ell}(\phi)^{\text{MC}} = -\sum_{i=1}^N \left[\int_{\mathbb{S}} \eta(s_{\text{dp}} \mid x^{(i)}) g(s_{\text{dp}}, \theta^{(i)}) \mathrm{d}s_{\text{dp}}\right]. \tag{10}$$

The inner integrals $I(\theta, x) = \int \eta(s_{\text{dp}} \mid x) g(s_{\text{dp}}, \theta) \mathrm{d}s_{\text{dp}}$ can be approximated with standard Monte Carlo integration techniques, as popular DP mechanisms such as Laplace and Gaussian mechanisms can be easily simulated. Using $M$ i.i.d. samples, the root-mean-squared-error (RMSE) to estimate $I(\theta, x)$ approximations are typically on the order of $\mathcal{O}(M^{-1/2})$ due to the central limit theorem.

In many applications, the DP query result $s_{\text{dp}}$ serves as a private descriptive statistic of a dataset $x$. This statistic is commonly embedded in a low-dimensional space $\mathbb{S}$, where the dimension $r = \dim(\mathbb{S})$ is significantly less than the dimension of the data space $\dim(\mathbb{X}^n)$. For the privacy mechanisms, we typically model the generation process as $\tau : (u, s(x)) \mapsto s_{\text{dp}}$ where $u \sim \mathcal{U}[0,1]^r$ is a uniform random variable from the $r$-dimensional hypercube. For example, the process $\tau(u, s(x)) = s(x) - \frac{\Delta_1(s)}{\epsilon} \text{sgn}(u - \frac{1}{2}) \log\left[1 - 2|u - \frac{1}{2}|\right]$ achieves $\epsilon-$DP for 1-dimensional queries.

Randomized quasi-Monte Carlo (RQMC) methods, as detailed in (Owen, 1997a;b), differ from traditional Monte Carlo (MC) methods in that it generate correlated, low-discrepancy sequences $\{v^{(1)}, \cdots, v^{(M)}\} \subset [0,1]^r$. These sequences cover the parameter space more evenly than the pseudo-random sequences used by MC

methods. This low-discrepancy feature helps to reduce the variance of the estimators, which is particularly useful in low-dimensional integration tasks (L'Ecuyer, 2018). The estimator used in RQMC can be described as

$$\hat{I}_{\theta,x}^{\mathrm{RQMC}} = \frac{1}{M} \sum_{j=1}^{M} g(\tau(v^{(j)}; x), \theta) := \frac{1}{M} \sum_{j=1}^{M} \tilde{g}_{\theta,x}(v^{(j)}). \tag{11}$$

The RQMC estimator is unbiased and has a smaller RMSE compared to MC estimators. The $\star$-discrepancy of the point set $\{v^{(1:M)}\}$, denoted by $D^\star(v^{(1:M)})$ is of the order $\mathcal{O}(M^{-1}(\log M)^r) = \mathcal{O}(M^{-1+\delta})$ for a positive constant $\delta$. If our neural approximation family $\tilde{g}_{\theta,x}(\cdot)$ has bounded Hardy-Krause variation $V_{\mathrm{HK}}[\tilde{g}_{\theta,x}(\cdot)]$ (Aistleitner et al., 2017), then, according to Basu & Owen (2016), the mean squared error of the RQMC estimator in Equation (11) satisfies

$$\mathbb{E}\left[\left(\hat{I}_{\theta,x}^{\mathrm{RQMC}} - I(\theta, x)\right)^2\right] \le V_{\mathrm{HK}}^2(\tilde{g}_{\theta,x})(D^\star)^2 = \mathcal{O}(M^{-2+2\delta}). \tag{12}$$

Thus, the RMSE of the RQMC estimator is of the order $\mathcal{O}(M^{-1+\delta})$, achieving faster convergence than an MC estimator with $\mathcal{O}(M^{-1/2})$. We verify this improvement in convergence from the RQMC estimator on the SIR model and linear regression example with the neural spline flow approximation family (Durkan et al., 2019), in Appendix E, Figure 7.

## 3.5 Sequential neural estimations

This section presents our sequential neural approximation methods on privacy-protected data. Our central goal is to approximate the private data posterior distribution $\pi(\theta \mid s_{\mathrm{dp}})$ given observed privatized data query $s_{\mathrm{dp}}$. We summarize the Sequential Private Posterior Estimation (SPPE) algorithm in Algo.1 and the Sequential Private Likelihood Estimation (SPLE) in Algo.2.

---

**Algorithm 1** Sequential private-data posterior estimation (SPPE)

---

**Input**: observed privatized summary statistics $s_{\mathrm{dp}}^o$, neural estimation family $q_\phi(\theta \mid s_{\mathrm{dp}})$, and confidential data simulator $f(x \mid \theta)$
**Initialization**: set $\tilde{p}_0(\theta) = \pi(\theta)$, simulated data filtration $\mathcal{D}_0 = \{\}$
**for** $r = 1, 2, \cdots, R$ **do**
  Sample $\{\theta^{(i)}\}_{i=1:N}$ from $\tilde{p}_{r-1}(\theta)$
  Simulate $x^{(i)} \sim f(\cdot \mid \theta^{(i)})$ for each $i$
  Update filtration $\mathcal{D}_r = \mathcal{D}_{r-1} \cup \{(\theta^{(i)}, x^{(i)})\}_{i=1:N}$
  Update $\phi \leftarrow \arg\min_\phi \hat{\ell}_{\mathrm{PPE-A}}(\phi)$ using $\hat{I}^{\mathrm{RQMC}}$ Equation (11) on $\mathcal{D}_r$
  Set proposal $\tilde{p}_r(\theta) = q_\phi(\theta \mid s_{\mathrm{dp}}^o)$
**end for**
**return** $\hat{\pi}(\theta \mid s_{\mathrm{dp}}^o) = q_\phi(\theta \mid s_{\mathrm{dp}}^o)$

---

Both SPPE and SPLE use normalizing flows as the variational family to minimize some KL divergence, which takes the general form of Equation (9). We have designed their sequential training procedures to be sample efficient, in the sense that training data generated during previous rounds are kept and used in subsequent rounds.

In a sequential approximation procedure, we iteratively refine the neural approximations towards the target distribution. After the $r$-th training round, we incorporate the current neural density estimator $q_\phi$ into the proposal distribution of the next training round, using the automatic posterior transformation weights and loss functions described in Equations (7) and (8) respectively. Sequential training procedures can gradually move $q_\phi$ towards high-density regions of the private data posterior, and thus achieve good accuracy with fewer samples from the simulator, compared with non-sequential training procedures that use a fixed proposal distribution.

---

**Algorithm 2** Sequential private-data likelihood estimation (SPLE)

---

**Input**: observed privatized summary statistics $s_{dp}^o$, neural estimation family $q_\phi(s_{dp} \mid \theta)$, and confidential data simulator $f(x \mid \theta)$
**Initialization**: set $\tilde{p}_0(\theta) := \pi(\theta)$, simulated data filtration $\mathcal{D}_0 = \{\}$
**for** $r = 1, 2, \cdots, R$ **do**
    Sample $\{\theta^{(i)}\}_{i=1}^N$ from $\tilde{p}_{r-1}(\theta)$.
    Simulate $x^{(i)} \sim f(\cdot \mid \theta^{(i)})$ for each $i$
    Update filtration $\mathcal{D}_r = \mathcal{D}_{r-1} \cup \{(\theta^{(i)}, x^{(i)})\}_{i=1:N}$
    Update $\phi \leftarrow \arg\min_\phi \hat{\ell}_{PLE}(\phi)$ using $\hat{I}^{RQMC}$ Equation (11) on $\mathcal{D}_r$
    Set proposal $\tilde{p}_r(\theta) \propto \pi(\theta) q_\phi(s_{dp}^o \mid \theta)$
**end for**
**return** posterior estimation $\widehat{\pi}(\theta \mid s_{dp}^o) = \tilde{p}_R(\theta)$ and likelihood estimation $\hat{f}_\theta(s_{dp} \mid \theta) = q_{\phi^\star}(s_{dp} \mid \theta)$

---

## 4 Applications

Here we illustrate our methods on the susceptible-infected-recovered (SIR) model and linear regression. We include experiments on the Naïve Bayes log-linear model in the Appendix.

### 4.1 SIR model for disease spread

The SIR model is a time-series model that describes how an infectious disease spreads in a closed population. It is most often used as a deterministic ordinary differential equation (ODE), but can also be represented by a Markov jump process.

To the best of our knowledge, inference on privacy-protected data with the SIR model has not been studied in the literature. Our proposed methods are particularly suitable for this problem for two reasons. First, our methods are simulation-based and thus are applicable under the ODE model, when other likelihood-based methods can no longer be applied. Second, in the SIR model, low-dimensional summary statistics can be very informative about model parameters. Then the RQMC methods discussed in Section 3.4 can provide efficiency and accuracy gains when evaluating the loss functions.

We describe a stochastic SIR model in a closed population with $K$ people. As the disease spreads, the individuals progress through the three states: susceptible, infected, and recovered. We use $S(t), I(t)$, and $R(t)$ to denote the number of individuals within each compartment at time $t$. We make the following assumptions: (a) individuals are infected at a rate $\beta \frac{SI}{K}$, resulting in a decrease of $S$ by one and an increase of $I$ by one, (b) infected individuals recover with a rate $\gamma I$, leading to a decrease of $I$ by one and an increase of $R$ by one. The confidential data likelihood of this continuous-time Markov jump process is hard to compute. Our goal is to infer the infection and recovery rates $\theta = (\beta, \gamma)$, under initial conditions $(S, I, R) = (K - 1, 1, 0)$.

**Privatizing the infection curve.** Here, we propose a mechanism to privatize the infection trajectory $I(t)/K$, which is the proportion of infected individuals at each $t$.

**Proposition 4** (DP infection trajectory). *Consider a sequence of $L$ points $\{t_1, \cdots, t_L\}$ in the time interval $[0, T]$, our privatized query can be $s_{dp} = (s_1, \cdots, s_L)$ where each $s_i \sim \text{Binomial}\left(n, \frac{I(t_i)+m}{K+2m}\right)$ independently. The mechanism generating $s_{dp} = (s_1, \cdots, s_L)$ satisfies $\epsilon$-DP, with $\epsilon = \frac{n}{m}L$.*

This algorithm adds calibrated noise to the SIR process to produce $s_{dp}$, a differential private time series. It can probably protect each individual's infection status. We demonstrate that analysts can still make inferences about population parameters $(\beta, \gamma)$ by only knowing $s_{dp}$, which retains information about the speed of disease spread.

**Experiments on synthetic data.** We illustrate the performance of SPPE and SPLE on synthetic privatized SIR model data. The data generating parameters are set to emulate a measles outbreak. We set $L = 10$ time points with $n = 1000$ and $m = 1000$, achieving $\epsilon = 10$ differential privacy. Lower privacy loss budgets

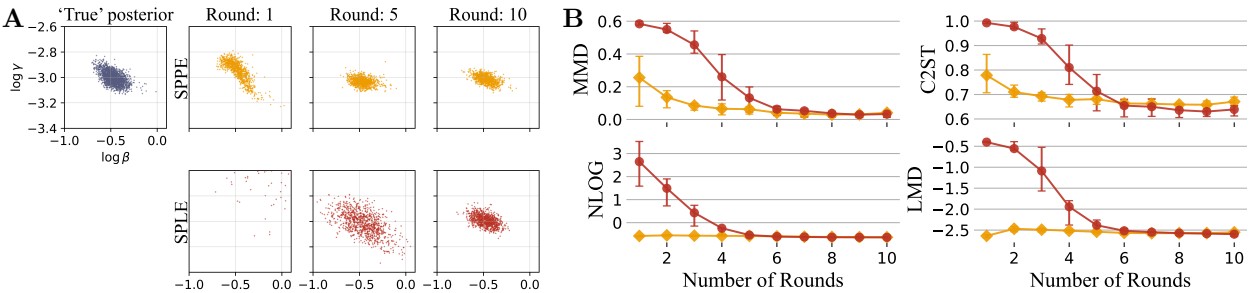

Figure 1: Inference on SIR model. **A.** Convergence of sequential posterior estimations given DP-protected infection trajectory. Each round entails $N = 1000$ simulations. **B.** Approximation accuracy by SPPE (orange) and SPLE (red) against the number of rounds, the error bars represent the mean with the upper and lower quartiles over 20 random trials.

($\epsilon = 1, 0.1$) are explored in Appendix E.1.1 by adjusting $m$ accordingly. We also describe prior specifications and implementation details in the Appendix.

Figure 1-A describes the convergence of the posterior approximations $\hat{\pi}(\theta \mid s_{\mathrm{dp}}^o)$ towards the SMC-ABC (Beaumont et al., 2009) baseline, requiring up to $5 \times 10^5$ simulations. We use SMC-ABC as the baseline because it does not resort to the variational approximations employed by SPPE and SPLE. Both SPPE and SPLE quickly adapt to meet the SMC-ABC results, with orders of magnitudes fewer simulations needed. After the first round of simulations, SPPE can identify the high probability region of $\pi(\theta \mid s_{\mathrm{dp}})$, while the SPLE posterior is still exploring the parameter space. See Appendix for the SPLE approximation after $r = 1$. After $r = 5$ rounds, both methods can concentrate around the posterior mean and have captured the posterior correlation $\mathrm{cov}(\beta, \gamma \mid s_{\mathrm{dp}})$. By inspecting marginal posterior histograms (Figure 6), we find that, in this example, the posterior approximated by SPPE is slightly more concentrated than those from SMC-ABC and SPLE.

To quantitatively evaluate the performance of our methods, we use the following metrics: (1) MMD (Gretton et al., 2012): maximum mean discrepancy between the neural estimated posteriors and SMC-ABC posterior; (2) C2ST (Lopez-Paz & Oquab, 2016): classifier two sample tests; (3) NLOG: negative log density at true parameters; and (4) LMD: log median distance from simulated to observed $s_{\mathrm{dp}}$. Smaller values indicate better performance for all four metrics. In Figure 1-B, we compare the performance of these methods at various numbers of simulation samples/rounds. After Round 5, SPLE and SPPE have similar accuracy. We also compare their runtime in Table 1 of the Appendix. To achieve MMD lower than 0.1, SPPE is 6x faster and SPLE is 2x faster than SMC-ABC.

**Real disease outbreaks.** We apply our privacy mechanism and inference methods to several real infectious disease outbreaks: influenza, Ebola, and COVID-19. In Figure 2-A, we compare the posterior distributions $\pi(\beta, \gamma \mid s_{\mathrm{dp}})$ obtained from SMC-ABC, SPPE, and SPLE. The results are similar to synthetic data experiments: when SPPE and SPLE use the same computational resources, the SPPE posterior converges faster than SPLE. In Figure 2-B, we inspect the 95%-credible interval for the ratio $R_0 = \beta/\gamma$, which is known as the basic reproduction number. Our $R_0$ estimates using privatized data are consistent with common estimates of $R_0$ for these diseases (Eisenberg, 2020). Specifically, typical $R_0$ ranges are $(1.51, 2.53)$ for Ebola, $(1.5, 3.5)$ for COVID-19, and with the exception of the flu outbreak $(0.9, 2.1)$, which should be modeled by the SI model instead of SIR.

## 4.2 Bayesian linear regression

We demonstrate our methods on a linear regression model, and compare it to existing work on DP regression analysis like (Ju et al., 2022; Bernstein & Sheldon, 2019). We consider linear regression with $n$ subjects and $p$ predictors. Denote $x_0 \in \mathbb{R}^{n \times p}$ as the design matrix without intercept terms, and let $x = (\mathbf{1}_{n \times 1}, x_0)$ represent the design matrix with the intercept. Ordinary linear regression models assume that the outcomes

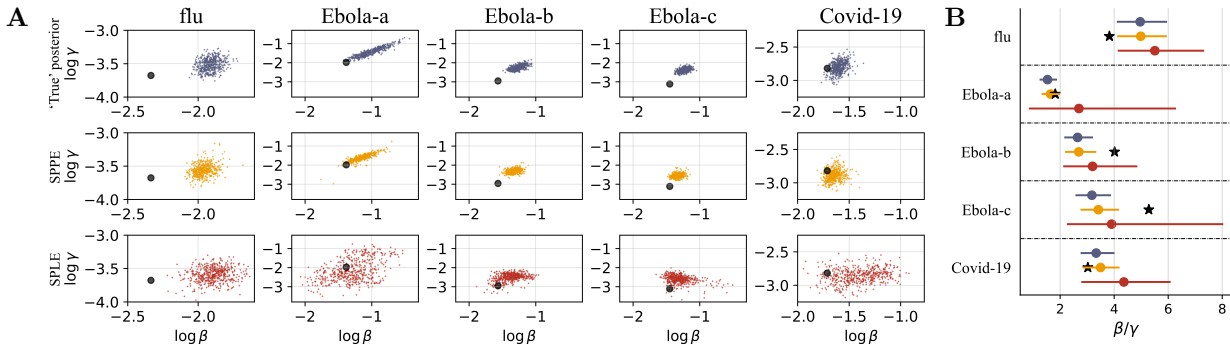

Figure 2: Inference on real infectious disease outbreaks. **A.** Visualization of the posterior distribution given private infection curve applied to flu, Ebola [in a) Guinea, b) Liberia, and c) Sierra Leone], and COVID-19 in Clark County, Nevada. All experiments use a privacy level of $\epsilon = 10$. **B.** Mean and 95% credible intervals for $R_0 = \beta/\gamma$ with different methods in each dataset. Grey: SMC-ABC; orange: SPPE; red: SPLE. A non-DP baseline (black stars) is also shown, which is obtained from the solution of the corresponding ordinary differential equations.

$y$ satisfy $y|x_0 \sim \mathcal{N}_n(x\beta, \sigma^2 I_n)$. Under the constraint of differential privacy, both outcomes $y$ and predictors $x_0$ are subject to calibrated noise. In a Bayesian setting, we model the predictors with $x_{0,i} \sim \mathcal{N}_p(m, \Sigma)$ for $i = 1, 2, \cdots, n$ independently. Our parameter of interest is $\beta$, which represents the $(p+1)$-dimensional vector of regression coefficients. Our experiments assume that $\sigma$, $m$, and $\Sigma$ are fixed, to illustrate our algorithm. In practice, one can also estimate these parameters from data. Our setting is the same as that used in Ju et al. (2022).

**Private sufficient statistics.** We achieve $\epsilon$-DP on confidential data $(x, y)$ by adding Laplace noise to sufficient statistics. Achieving privacy requires finite $\ell_1$ sensitivity on confidential data. As a result, before adding noise for privacy, we first need to clamp the predictors and the responses, and then normalize them to take values in $[-1, 1]$. Let's denote the clamped confidential data as $\tilde{x}$ and $\tilde{y}$, respectively. We then define the summary statistics of clamped data as $\tilde{s} := \left(\frac{1}{n}\tilde{x}^\top \tilde{y}, \frac{1}{n}\tilde{y}^\top \tilde{y}, \frac{1}{n}\tilde{x}^\top \tilde{x}\right)$. We have refined the sensitivity analysis of Ju et al. (2022) to $\Delta(s) = \frac{1}{n}(p^2 + 3p + 3)$. The privacy summary statistics $s_{\mathrm{dp}}$ is achieved by adding independent Laplace$(0, \Delta_1(\tilde{s})/\epsilon)$ noise to each entry of $\tilde{s}$. This output perturbation mechanism satisfies $\epsilon$-DP. Details of clamping and sensitivity analysis are in the supplementary materials.

**Posterior approximations.** We compare the 95% posterior credible intervals obtained by methods applicable to linear regression, including SMC-ABC, SPPE, SPLE, Data-augmentation MCMC (DA-MCMC) (Ju et al., 2022), Gibbs-SS (Bernstein & Sheldon, 2019), and RNPE (Ward et al., 2022). See Table 2, 3, and 4 for marginal posterior credible intervals of $\beta \mid s_{\mathrm{dp}}$ at privacy levels $\epsilon = 10$, 1 and 0.1 respectively. Both tables are in Appendix E.2. We also use the Kolmogorov-Smirnov test to assess the similarity of empirical posterior marginal, shown in Figure 3 at a privacy level of $\epsilon = 10$. Results from SPPE, SPLE, SMC-ABC, and DA-MCMC reach agreement on the posterior marginals. However, results from Gibbs-SS and RNPE are qualitatively different from the other four methods, yielding biased approximations for $\beta_0$ and $\beta_1$ respectively. Among the likelihood-free methods, SPLE attempts to approximate the likelihood function and is the most similar to DA-MCMC, our likelihood-based baseline.

**The cost of privacy.** Although it has been a standard practice in many DP work (Bernstein & Sheldon, 2018; 2019; Ju et al., 2022; Gong, 2022) to achieve finite global sensitivity through clamping, this benefit of privacy protection comes at the cost of accuracy of the subsequent statistical analysis.

Our experiments demonstrate the privacy-utility trade-off, which has also been demonstrated by related works on noise/privacy-aware inference. A naive plug-in estimator (plugging in $s_{\mathrm{dp}}$ as $s$ into the conjugate posterior $\pi(\theta \mid s)$) gives the wrong posterior; See the second rows in Table 2. Besides, achieving privacy protection comes at the cost of estimation accuracy: private data posterior $\pi(\theta \mid s_{\mathrm{dp}})$ is different from the

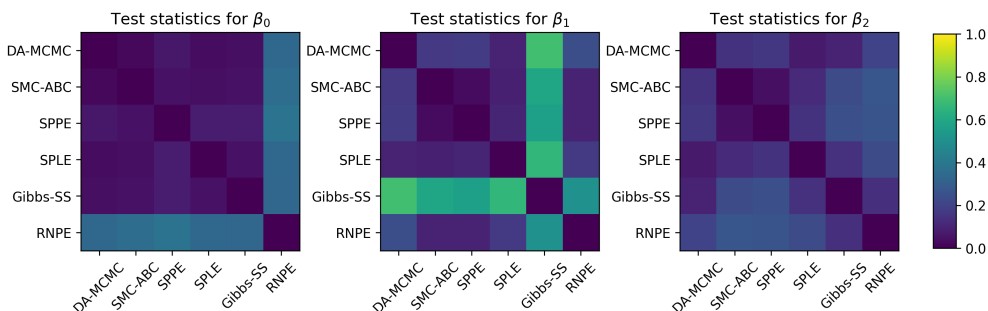

Figure 3: Kolmogorov-Smirnov test statistics between approximations of posterior marginals, at $\epsilon = 10$.

confidential data posterior $\pi(\theta \mid x)$ even under a high loss budget of $\epsilon = 10$ (small privacy noise) setting; See the first rows in Table 2, 3, and 4. With small privacy noise, data corruption primarily comes from the clamping step. Evaluating this censoring bias is still an open problem in DP data analysis, with some recent attempts in Biswas et al. (2020); Evans et al. (2019); Covington et al. (2021).

## 5 Discussion

In this work, we propose three simulation-based inference methods to learn population parameters from privacy-protected data: SMC-ABC, SPPE, and SPLE. The latter two are neural density estimation methods. We have designed SPPE and SPLE to leverage state-of-the-art computational tools, such as normalizing flows and randomized quasi-Monte Carlo, to be computationally efficient for complex data models. SPPE aims to approximate the posterior-data posterior, and SPLE approximates the posterior-data marginal likelihood.

We compare our methods to similar DP data analysis work that focuses on *post processing* of privacy-protected datasets (or their summary statistics). Compared with existing ABC-based analysis for DP data (Gong, 2022), SPPE and SPLE do not reject training samples and hence are more computationally efficient. Compared with DA-MCMC (Ju et al., 2022), our method does not require that the confidential data likelihood can be evaluated easily. Our methods require only that one can simulate from the prior distribution $\pi(\theta)$ and the confidential model $f(x \mid \theta)$. They also all scale linearly with the sample size of the confidential database.

Our work contributes to the growing literature on statistical analysis with privatized data. In particular, our simulation-based inference framework can be applied to complex models with intractable likelihood functions and the resulting triply-intractable private data posterior. We hope our methods can catalyze DP-protected data sharing between data collectors and analysts. Our experiments and analysis demonstrate the necessity and feasibility of designing valid statistical inference procedures to correct for biases introduced by privacy-protection mechanisms. We advocate for increases in both sharing privacy-protected data by collectors and using valid inference procedures.

We acknowledge the limitations of the present work and point out future directions. Our methods leverage the fact that popular DP mechanisms can be efficiently achieved with random number generators. In some tasks, such as DP principle component analysis (Chaudhuri et al., 2013), the privacy mechanism is actually intractable to simulate from but its density is easy to evaluate. Our method is not applicable to this type of DP mechanism. Additionally, since our method scales linearly in the sample size of the confidential database, it might not be ideal for massive datasets, such as the Facebook URL dataset (Evans & King, 2023), which concerns millions of users. It is of interest to develop methods that scale sub-linearly in sample size. Finally, in Appendix D, we show the algorithms scale linearly with the dimension of the summary statistics and quadratically with the number of nodes per layer in the neural network. In our experiments, we found that using at most 8 layers and 50 nodes can lead to satisfactory performance. A precise understanding of how the convergence rate depends on the sample size of the confidential data and the dimensionality of the summary statistics remains to be established and is likely to vary on a case-by-case basis for the choice neural density estimators.

## Acknowledgments

We sincerely appreciate the valuable comments and suggestions from anonymous referees. The work of the third author was supported by the National Natural Science Foundation of China No.12171454 and Fundamental Research Funds for the Central Universities.

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

## A   Derivations for SPLE and SPPE objectives

**Lemma 5.** *For private data likelihood estimation, minimizing the average KL divergence* $\mathcal{D}_{\mathrm{KL}}\left(f(s_{\mathrm{dp}} \mid \theta)\|q_\phi(s_{\mathrm{dp}} \mid \theta)\right)$ *under the prior* $\pi(\theta)$*, is equivalent to minimizing the objective function*

$$\ell_{\mathrm{PLE}}(\phi) = \mathbb{E}_{p(\theta,x)}\left[-\int_{\mathbb{S}} \eta(s_{\mathrm{dp}} \mid x)\log q_\phi(s_{\mathrm{dp}} \mid \theta)\mathrm{d}s_{\mathrm{dp}}\right]. \tag{13}$$

*With respect to the joint distribution* $\tilde{p}(\theta, x) \propto \tilde{p}(\theta)f(x \mid \theta)$*, the objective function still has the form*

$$\ell_{\mathrm{PLE}}(\phi) = \mathbb{E}_{\tilde{p}(\theta,x)}\left[-\int_{\mathbb{S}} \eta(s_{\mathrm{dp}} \mid x)\log q_\phi(s_{\mathrm{dp}} \mid \theta)\mathrm{d}s_{\mathrm{dp}}\right]. \tag{14}$$

*Proof.* Note that

$$
\begin{aligned}
&\mathbb{E}_{\pi(\theta)}\left[\mathcal{D}_{\mathrm{KL}}\left(f(s_{\mathrm{dp}} \mid \theta)\|q_\phi(s_{\mathrm{dp}} \mid \theta))\right)\right] \\
&= \int_\Theta \pi(\theta)\left[\int_\mathbb{S} f(s_{\mathrm{dp}} \mid \theta)\left(\log f(s_{\mathrm{dp}} \mid \theta) - \log q_\phi(s_{\mathrm{dp}} \mid \theta)\right) \mathrm{d}s_{\mathrm{dp}}\right] \mathrm{d}\theta \\
&= C_1 - \iint_{\mathbb{S}\times\Theta} \pi(\theta)f(s_{\mathrm{dp}} \mid \theta)\log q_\phi(s_{\mathrm{dp}} \mid \theta)\mathrm{d}s_{\mathrm{dp}}\mathrm{d}\theta \\
&= C_1 - \iiint_{\mathbb{S}\times\Theta\times\mathbb{X}^n} \pi(\theta)\eta(s_{\mathrm{dp}} \mid x)f(x \mid \theta)\log q_\phi(s_{\mathrm{dp}} \mid \theta)\mathrm{d}s_{\mathrm{dp}}\mathrm{d}\theta\mathrm{d}x \\
&= C_1 - \iint_{\Theta\times\mathbb{X}^n} p(\theta,x)\left[\int_\mathbb{S} \eta(s_{\mathrm{dp}} \mid x)\log q_\phi(s_{\mathrm{dp}} \mid \theta)\mathrm{d}s_{\mathrm{dp}}\right] \mathrm{d}\theta\mathrm{d}x \\
&= C_1 - \mathbb{E}_{p(\theta,x)}\left[\int_\mathbb{S} \eta(s_{\mathrm{dp}} \mid x)\log q_\phi(s_{\mathrm{dp}} \mid \theta)\mathrm{d}s_{\mathrm{dp}}\right],
\end{aligned}
$$

where $C_1 = \iint_{\mathbb{S}\times\Theta} \pi(\theta)f(s_{\mathrm{dp}} \mid \theta)\log f(s_{\mathrm{dp}} \mid \theta)\mathrm{d}s_{\mathrm{dp}}\mathrm{d}\theta$ is a constant unrelated to $\phi$. Furthermore, if $\theta$ are sampled from some proposal $\tilde{p}(\theta)$, we still have

$$
\begin{aligned}
&\mathbb{E}_{\tilde{p}(\theta)}\left[\mathcal{D}_{\mathrm{KL}}\left(f(s_{\mathrm{dp}} \mid \theta)\|q_\phi(s_{\mathrm{dp}} \mid \theta))\right)\right] \\
&= \iint_{\mathbb{S}\times\Theta} \tilde{p}(\theta)f(s_{\mathrm{dp}} \mid \theta)\log f(s_{\mathrm{dp}} \mid \theta)\mathrm{d}s_{\mathrm{dp}}\mathrm{d}\theta - \mathbb{E}_{\tilde{p}(\theta,x)}\left[\int_\mathbb{S} \eta(s_{\mathrm{dp}} \mid x)\log q_\phi(s_{\mathrm{dp}} \mid \theta)\mathrm{d}s_{\mathrm{dp}}\right].
\end{aligned}
$$

$\square$

**Lemma 6.** *For private data posterior estimation, minimizing the average KL divergence*
$\mathcal{D}_{\mathrm{KL}}\left(p(\theta \mid s_{\mathrm{dp}})\|q_\phi(\theta \mid s_{\mathrm{dp}}))\right)$ *with respect to the marginal evidence* $p(s_{\mathrm{dp}}) = \int_\mathbb{S} \pi(\theta)f(s_{\mathrm{dp}} \mid \theta)\mathrm{d}\theta$, *is equivalent to minimizing the objective function*

$$
\ell_{\mathrm{PPE}}(\phi) = \mathbb{E}_{p(\theta,x)}\left[-\int_\mathbb{S} \eta(s_{\mathrm{dp}} \mid x)\log q_\phi(\theta \mid s_{\mathrm{dp}})\mathrm{d}s_{\mathrm{dp}}\right]. \tag{15}
$$

*With respect to the joint distribution* $\tilde{p}(\theta,x) \propto \tilde{p}(\theta)f(x \mid \theta)$, *then the objective function has the form*

$$
\ell_{\mathrm{PPE-A}}(\phi) = \mathbb{E}_{\tilde{p}(\theta,x)}\left[-\int_\mathbb{S} \eta(s_{\mathrm{dp}} \mid x)\log \tilde{q}_\phi(\theta \mid s_{\mathrm{dp}})\mathrm{d}s_{\mathrm{dp}}\right], \tag{16}
$$

*where*

$$
\tilde{q}_\phi(\theta \mid s_{\mathrm{dp}}) := q_\phi(\theta \mid s_{\mathrm{dp}})\frac{\tilde{p}(\theta)}{\pi(\theta)}\frac{1}{Z(s_{\mathrm{dp}},\phi)}, \quad Z(s_{\mathrm{dp}},\phi) = \int_\Theta q_\phi(\theta \mid s_{\mathrm{dp}})\frac{\tilde{p}(\theta)}{\pi(\theta)}\mathrm{d}\theta. \tag{17}
$$

*Proof.* We have

$$
\begin{aligned}
&\mathbb{E}_{p(s_{\mathrm{dp}})}\left[\mathcal{D}_{\mathrm{KL}}\left(\pi(\theta \mid s_{\mathrm{dp}})\|q_\phi(\theta \mid s_{\mathrm{dp}}))\right)\right] \\
&= \int_\mathbb{S} p(s_{\mathrm{dp}})\left[\int_\Theta \pi(\theta \mid s_{\mathrm{dp}})\left(\log \pi(\theta \mid s_{\mathrm{dp}}) - \log q_\phi(\theta \mid s_{\mathrm{dp}})\right) \mathrm{d}\theta\right] \mathrm{d}s_{\mathrm{dp}} \\
&= C_2 - \iint_{\mathbb{S}\times\Theta} p(s_{\mathrm{dp}})\pi(\theta \mid s_{\mathrm{dp}})\log q_\phi(\theta \mid s_{\mathrm{dp}})\mathrm{d}s_{\mathrm{dp}}\mathrm{d}\theta \\
&= C_2 - \iiint_{\mathbb{S}\times\Theta\times\mathbb{X}^n} \pi(\theta)\eta(s_{\mathrm{dp}} \mid x)f(x \mid \theta)\log q_\phi(\theta \mid s_{\mathrm{dp}})\mathrm{d}s_{\mathrm{dp}}\mathrm{d}\theta\mathrm{d}x \\
&= C_2 - \iint_{\Theta\times\mathbb{X}^n} p(\theta,x)\left[\int_\mathbb{S} \eta(s_{\mathrm{dp}} \mid x)\log q_\phi(\theta \mid s_{\mathrm{dp}})\mathrm{d}s_{\mathrm{dp}}\right] \mathrm{d}\theta\mathrm{d}x \\
&= C_2 - \mathbb{E}_{p(\theta,x)}\left[\int_\mathbb{S} \eta(s_{\mathrm{dp}} \mid x)\log q_\phi(\theta \mid s_{\mathrm{dp}})\mathrm{d}s_{\mathrm{dp}}\right],
\end{aligned}
$$

where $C_2 = \iint_{\mathbb{S} \times \Theta} p(s_{\mathrm{dp}}) \pi(\theta \mid s_{\mathrm{dp}}) \log \pi(\theta \mid s_{\mathrm{dp}}) \mathrm{d}s_{\mathrm{dp}} \mathrm{d}\theta$ is a constant unrelated to $\phi$. Furthermore, if $\theta$ are sampled from some proposal $\tilde{p}(\theta)$, then

$$
\mathbb{E}_{p(s_{\mathrm{dp}})} \left[ \mathcal{D}_{\mathrm{KL}} \left( \tilde{\pi}(\theta \mid s_{\mathrm{dp}}) \| \tilde{q}_\phi(\theta \mid s_{\mathrm{dp}}) \right) \right]
$$
$$
= \iint_{\mathbb{S} \times \Theta} p(s_{\mathrm{dp}}) \tilde{\pi}(\theta \mid s_{\mathrm{dp}}) \log \tilde{\pi}(\theta \mid s_{\mathrm{dp}}) \mathrm{d}s_{\mathrm{dp}} \mathrm{d}\theta - \mathbb{E}_{\tilde{p}(\theta,x)} \left[ \int_{\mathbb{S}} \eta(s_{\mathrm{dp}} \mid x) \log \tilde{q}_\phi(\theta \mid s_{\mathrm{dp}}) \mathrm{d}s_{\mathrm{dp}} \right], \tag{18}
$$

where $\tilde{\pi}(\theta \mid s_{\mathrm{dp}})$ is called *proposal posterior* (Greenberg et al., 2019), which satisfied

$$
\tilde{\pi}(\theta \mid s_{\mathrm{dp}}) = \pi(\theta \mid s_{\mathrm{dp}}) \frac{\tilde{p}(\theta) p(s_{\mathrm{dp}})}{\pi(\theta) \tilde{p}(s_{\mathrm{dp}})}, \quad \tilde{p}(s_{\mathrm{dp}}) = \int_\Theta \tilde{p}(\theta) f(s_{\mathrm{dp}} \mid \theta) \mathrm{d}\theta.
$$

Based on Prop. 1 in the work of Papamakarios & Murray (2016), minimizing Equation (18) results in the convergence of $\tilde{q}_\phi(\theta \mid s_{\mathrm{dp}})$ to $\tilde{\pi}(\theta \mid s_{\mathrm{dp}})$ and $q_\phi(\theta \mid s_{\mathrm{dp}})$ to $\pi(\theta \mid s_{\mathrm{dp}})$. $\qquad \square$

## B  Proof of Proposition 4

**Proposition 4.** *Consider a sequence of $L$ points $\{t_1, \cdots, t_L\}$ in the time interval $[0, T]$, our privatized query can be $s_{\mathrm{dp}} = (s_1, \cdots, s_L)$ where each $s_i \sim \mathrm{Binomial}\left(n, \frac{I(t_i)+m}{K+2m}\right)$ independently. The mechanism generating $s_{\mathrm{dp}} = (s_1, \cdots, s_L)$ satisfies $\epsilon$-DP, with $\epsilon = \frac{n}{m} L$.*

*Proof.* Denote the numbers of infectious as $I(t) \in \{0, 1, \cdots, K\}$ and its neighbors $\tilde{I}(t)$, note that $|I(t) - \tilde{I}(t)| \leq 1$ holds for all $t \in [0, T]$ because a change of the infection status of any one of the $K$ individuals will at most increase or decrease $I(t)$ by only 1. We examine the following density ratio:

$$
\frac{p(s_i \mid I(t_i))}{p(s_i \mid \tilde{I}(t_i))} = \frac{\binom{n}{s_i} \left(\frac{I(t_i)+m}{K+2m}\right)^{s_i} \left(\frac{K-I(t_i)+m}{K+2m}\right)^{(n-s_i)}}{\binom{n}{s_i} \left(\frac{\tilde{I}(t_i)+m}{K+2m}\right)^{s_i} \left(\frac{K-\tilde{I}(t_i)+m}{K+2m}\right)^{(n-s_i)}}
$$
$$
= \left(\frac{I(t_i)+m}{\tilde{I}(t_i)+m}\right)^{s_i} \left(\frac{K-I(t_i)+m}{K-\tilde{I}(t_i)+m}\right)^{(n-s_i)}
$$
$$
:= H_i \ .
$$

This expression can be analyzed under three distinct cases. In the first case, when $\tilde{I}(t_i) > I(t_i)$, we have $\tilde{I}(t_i) = I(t_i) + 1$, leading to $H_i \leq \left(\frac{K-I(t_i)+m}{K-\tilde{I}(t_i)+m}\right)^{(n-s_i)} \leq \left(\frac{K-I(t_i)+m}{K-\tilde{I}(t_i)+m}\right)^n \leq \left(\frac{1+m}{m}\right)^n$. In the second case, if $\tilde{I}(t_i) < I(t_i)$, then $\tilde{I}(t_i) = I(t_i) - 1$, and we obtain $H_i \leq \left(\frac{I(t_i)+m}{\tilde{I}(t_i)+m}\right)^{s_i} \leq \left(\frac{I(t_i)+m}{\tilde{I}(t_i)+m}\right)^n \leq \left(\frac{1+m}{m}\right)^n$. Lastly, when $\tilde{I}(t_i) = I(t_i)$, $H_i$ equals to 1. Thus, $\frac{p(s_i \mid I(t_i))}{p(s_i \mid \tilde{I}(t_i))} \leq \left(\frac{1+m}{m}\right)^n \leq \exp\left(\frac{n}{m}\right)$. Now for $s_{\mathrm{dp}} = (s_1, \cdots, s_L)$, we have

$$
\frac{p(s_{\mathrm{dp}} \mid \{I(t) | t \in [0, T]\})}{p(s_{\mathrm{dp}} \mid \{\tilde{I}(t) | t \in [0, T]\})} = \frac{p(s_{\mathrm{dp}} \mid I(t_1), \cdots, I(t_L))}{p(s_{\mathrm{dp}} \mid \tilde{I}(t_1), \cdots, \tilde{I}(t_L))} \leq \exp\left(\frac{n}{m} L\right),
$$

which gives the mechanism generating $s_{\mathrm{dp}} = (s_1, \cdots, s_L)$ satisfies $\epsilon$-DP, with $\epsilon = \frac{n}{m} L$. $\qquad \square$

## C  Description of the SMC-ABC algorithm

Algorithm 3 presents the extension of the SMC-ABC algorithm (Sisson et al., 2007; Beaumont et al., 2009) to posterior estimation from the privatized data settings, by incorporating the private data simulator $f(x \mid \theta)$ and the privatized algorithm $\eta$.

---

**Algorithm 3** Sequential Monte Carlo - Approximate Bayesian Computation (SMC-ABC)

---

**Input**: observed privatized summary statistics $s_{\mathrm{dp}}^o$, discrepancy function $\rho$, thresholds $\{\epsilon_t\}$, transition kernels $\{K_t\}$, simulator $f(x \mid \theta)$, and privatized algorithm $\eta$

**for** $t = 1, 2, \ldots, T$ **do**
    **for** $i = 1, 2, \ldots, N$ **do**
        **if** $t = 1$ **then**
            Sample $\theta^{**} \sim \pi(\theta)$ independently
        **else**
            Sample $\theta^*$ from the previous population $\{\theta_{t-1}^{(i)}\}$ with weights $\{W_{t-1}^{(i)}\}$
            Perturb $\theta^*$ using transition kernel: $\theta^{**} \sim K_t(\theta \mid \theta^*)$
        **end if**
        **repeat**
            Generate $x^{**} \sim f(x \mid \theta^{**})$
            Generate $s_{\mathrm{dp}}^{**} \sim \eta(s_{\mathrm{dp}} \mid x^{**})$
        **until** $\rho(s_{\mathrm{dp}}^{**}, s_{\mathrm{dp}}^o) \leq \epsilon_t$
        Set $\theta_t^{(i)} = \theta^{**}$
        Compute weights:
$$W_t^{(i)} = \begin{cases} 1, & \text{if } t = 1 \\ \frac{\pi(\theta_t^{(i)})}{\sum_{j=1}^N W_{t-1}^{(j)} K_t(\theta_t^{(i)} \mid \theta_{t-1}^{(j)})}, & \text{if } t > 1 \end{cases}$$

    **end for**
    Normalize the weights so that $\sum_{i=1}^N W_t^{(i)} = 1$
    Compute effective sample size: $ESS = \left[ \sum_{i=1}^N (W_t^{(i)})^2 \right]^{-1}$
    **if** $ESS < E$ **then**
        Resample $\{\theta_t^{(i)}\}$ with weights $\{W_t^{(i)}\}$ to obtain a new population
        Set weights $\{W_t^{(i)} = 1/N\}$
    **end if**
**end for**
**return** posterior estimation $\{\theta_T^{(i)}, W_T^{(i)}\}_{i=1}^N$

---

# D Computational Complexity Analysis

In this section, we analyze the computational complexity of the proposed (S)PLE and (S)PPE algorithms in detail. We begin by examining the computational cost of neural spline flows, which serve as the core density estimators in our framework.

**Computational complexity for neural spline flows.** Denote the neural spline flow (Durkan et al., 2019) as $q_\phi(y \mid z)$ with trainable parameters $\phi$. Let $n_l$ represent the number of flow layers, and each layer containing $n_h$ hidden layers of $n_u$ units to generate the parameters of $K$ knots for monotonic spline. Assuming the input dimensions of $y$ and $z$ are $d_y$ and $d_z$, respectively. Each flow layer processes the concatenated input $(y, z)$ and outputs $(3K - 1)$ parameter for constructing the monotonic spline for each dimension of $y$ (see Appendix A.1 in Durkan et al. (2019) for details).

The computational complexity of evaluating gradient on one sample, i.e., $\nabla_\phi q_\phi(y_i \mid z_i)$, is then given by

$$C_{\mathrm{NSF}} = \mathcal{O}\left( n_l \left[ (d_z + d_y) n_u + n_h n_u^2 + K d_y n_u \right] \right). \tag{19}$$

**Computational complexity for PLE.** To minimize the PLE loss $\ell_{\mathrm{PLE}}(\phi)$, the gradient with respect to $\phi$ is computed over mini-batches of size $B$. For each pair $(\theta_i, x_i)$ in a mini-batch, the integral $I(\theta_i, x_i)$ from Equation (12) is estimated using $M$ pseudo-samples $\{v^{(1)}, \ldots, v^{(M)}\}$ generated via RQMC.

Considering a neural spline flow with $n_l$ layers, each containing $n_h$ hidden layers with $n_u$ units, the computational cost for processing a single mini-batch is then

$$C_{\text{PLE}} = \mathcal{O}\left(n_l BM\left[(\dim(\theta) + \dim(s))n_u + n_h n_u^2 + K\dim(s)n_u\right] + n\dim(\mathbb{X})BM\log M\right), \qquad (20)$$

where the last term represents the complexity of generating an RQMC-based estimator with $M$ samples. Specifically, the factor $M\log M$ accounts for the cost of generating an RQMC sequence, and $\dim(x) = n\dim(\mathbb{X})$ represents the dimension of the confidential data $x \in \mathbb{X}^n$ with sample size $n$.

**Computational complexity for PPE.** For the PPE method, the roles of the parameters $\theta$ and summary statistics $s_{\text{dp}}$ are reversed compared to PLE. To estimate the normalization constant $Z(s_{\text{dp}}, \phi)$ in Equation (17), we using the atomic proposal method (Greenberg et al., 2019), approximated as

$$Z(s_{\text{dp}}, \phi) = \int_{\Theta} q_\phi(\theta \mid s_{\text{dp}})\frac{\tilde{p}(\theta)}{\pi(\theta)}\mathrm{d}\theta \approx \frac{1}{n_a}\sum_{j=1}^{n_a} q_\phi(\theta_j \mid s_{\text{dp},j})\frac{1}{\pi(\theta_j)},$$

where $n_a$ is the number of atoms, and $\{\theta_j\}_{j=1}^{n_a}$ are subsamples under the current mini-batch, which can be regarded as the samples from proposal $\tilde{p}(\theta)$ (see Appendix A.2 in Greenberg et al. (2019) for details). The computational complexity for PPE is then given by

$$C_{\text{PPE}} = \mathcal{O}\left(n_l BM\left[(\dim(\theta) + \dim(s))n_u + n_a n_h n_u^2 + K\dim(s)n_a n_u\right] + n\dim(\mathbb{X})BM\log M\right). \qquad (21)$$

Both PLE and PPE scale linearly with the posterior dimension $\dim(\theta)$. Notably, while they also scale linearly with the confidential data sample size $n$, this scaling is decoupled from the complexity of the flow structure, which directly models the conditional density between the summary statistics $s$ (or $s_{\text{dp}}$) and the model parameters $\theta$, rather than the high-dimensional confidential data itself.

# E    Experimental details

The training and inference processes of the methods were primarily implemented using the Pytorch package in Python.

**Experimental setup.** We employed neural spline flows (Durkan et al., 2019) as the conditional density estimator, consisting of 8 layers. Following the flow structure settings in (Lueckmann et al., 2021), each layer consists of two residual blocks with 50 units and ReLU activation function, with 10 bins in each monotonic piecewise rational-quadratic transforms, and the tail bound was set to 5. Through empirical evaluation, we found that 8 layers provided sufficient flexibility for posterior estimation. Appendix E.4 compares 5-layer and 10-layer models, showing that 5 layers slightly degrade performance, while 10 layers perform similarly to 8 layers but with higher computational cost.

In the training process, the number of samples simulated in each round is $N = 1000$ and there are $R = 10$ rounds in total. In each round of training, we randomly select 5% of the newly generated samples as validation data. According to the early stop criterion proposed by Papamakarios et al. (2019), we stop training if the value of loss on validation data does not decrease after 20 epochs in a single round. For stochastic gradient descent optimizer, we choose the Adam (Kingma & Ba, 2014) with the batchsize of 100, the learning rate of $5 \times 10^{-4}$ and the weight decay is $10^{-4}$.

## E.1    SIR model

### E.1.1    Detailed results on synthetic data

In our experiments, we use the Gillespie algorithm (Gillespie, 1977) to simulate the whole process over a duration of $T = 160$ time units and record the populations at intervals of 1-time units. The prior distribution of $\beta$ is set to $\mathcal{N}(\log 0.4, 0.5)$ and prior distribution of $\gamma$ is set to $\mathcal{N}(\log 0.125, 0.2)$. The value of $K$ is configured as 1000000, while $N$ is set to 1000 for the number of observations.

To publish the privatized data about the infection process, we select the infectious group $I(t)$ at $L = 10$ evenly-spaced points in time $[0, T]$, with the privacy parameters set to $n = 1000$ and $m = 1000$, which satisfied $\epsilon$-DP with $\epsilon = 10$. The ground truth parameters are

$$\theta^* = (\exp(-0.5), \ \exp(-3)),$$

and the observed private statistic $s_{\mathrm{dp}}^o$ simulated from the model with ground truth parameters $\theta^*$ is

$$s_{\mathrm{dp}}^o = (0.0010, \ 0.0310, \ 0.6140, \ 0.2630, \ 0.1230, \ 0.0470, \ 0.0180, \ 0.0090, \ 0.0050, \ 0.0030).$$

Figure 4 illustrates the convergence of the approximate posterior by SPPE and SPLE in each round, and compares our results with the SMC-ABC method, where we performed simulations up to $5 \times 10^5$ times for the SMC-ABC method to generate the near exact 'True' posterior. The tolerance parameter $\epsilon_t$ in SMC-ABC was selected as $\epsilon_t = 0.5 \times 0.7^t$, where $t$ represents the iteration round, and the discrepancy function $\rho$ was chosen as the $\ell_2$ norm. Figure 5 depicts the results of the SMC-ABC method under the same performance metrics. Our method, after 10 rounds or $10^4$ simulations, achieved a similar performance as the SMC-ABC method with approximately $10^5$ simulations. The computational time costs comparison between our methods and SMC-ABC, as illustrated in Table 1, also reveals that our approaches require significantly fewer simulations, resulting in substantially lower simulation time expenditures compared to the SMC-ABC method. However, our methods require extra time for the training of normalizing flows, a duration dependent on the flow's complexity. Overall, both SPPE and SPLE attain a low MMD more rapidly than SMC-ABC.

Table 1: Computational Cost to achieve MMD $< 0.1$ in the SIR Model (Mean $\pm$ Standard Deviation)

| Method | Simulation Time (min) | Network Training Time (min) | Total | Number of Simulations |
|---|---|---|---|---|
| SMC-ABC | $115.38 \pm 1.51$ | - | $115.38 \pm 1.51$ | $82.85 \pm 1.03$ |
| SPPE | $\mathbf{2.65 \pm 1.03}$ | $\mathbf{15.73 \pm 7.67}$ | $\mathbf{18.38 \pm 8.63}$ | $\mathbf{2.12 \pm 0.71}$ |
| SPLE | $7.11 \pm 1.01$ | $45.98 \pm 14.46$ | $53.09 \pm 15.26$ | $5.40 \pm 0.77$ |

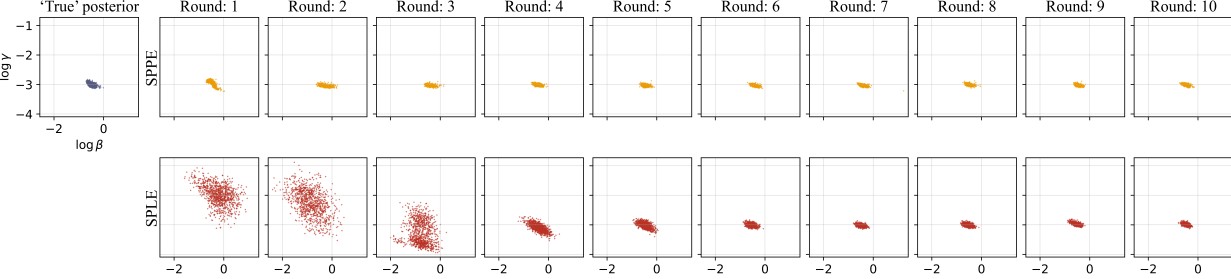

Figure 4: Detailed convergence of sequential posterior estimations given DP-protected infection trajectory under the SIR model. Each round entails $N = 1000$ simulations. Orange: SPPE; red: SPLE; grey: SMC-ABC.

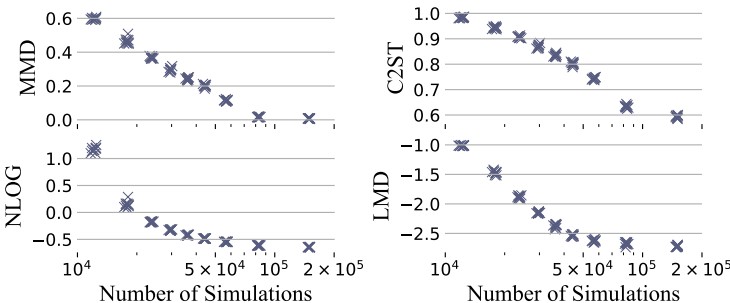

Figure 5: Approximation accuracy by SMC-ABC on the SIR model against the number of simulations.

Finally, we investigate the marginal posterior distributions $\pi(\beta \mid s_{\mathrm{dp}})$ and $\pi(\gamma \mid s_{\mathrm{dp}})$ in Figure 6, and conclude that SPPE and SPLE perform similarly well. In this example, the posterior approximated by SPPE is slightly more concentrated than those from SMC-ABC and SPLE.

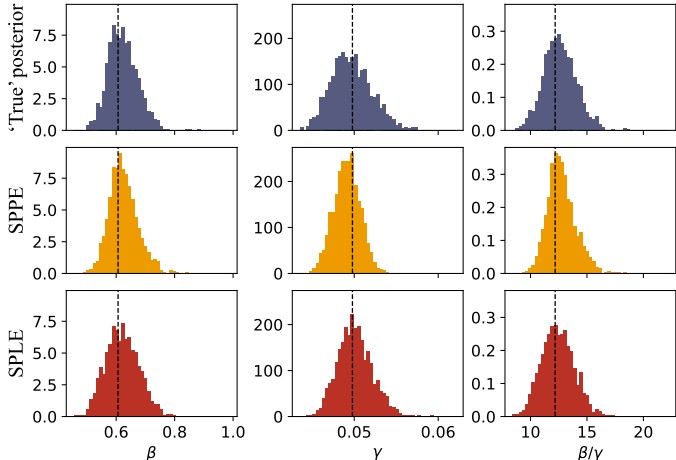

Figure 6: Marginal posterior histograms of $\beta, \gamma$, and $\beta/\gamma$ in SIR model on synthetic data. Grey: SMC-ABC; orange: SPPE; red: SPLE. The vertical lines indicate true data generating parameters, set to mimic a measles outbreak.

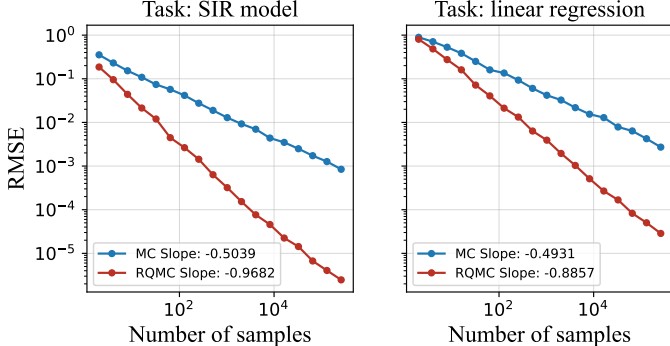

Figure 7: Rate of convergence of MC and RQMC: The $x$-axis represents the number of samples $M$ used for integral estimation, and the $y$-axis shows the logarithmic value of the root-mean-square-error (RMSE). The results indicate that the RMSE of the MC method is approximately $\mathcal{O}(M^{-1/2})$, while the RMSE of the RQMC method is approximately $\mathcal{O}(M^{-1})$.

Finally, we evaluate the impact of the privacy loss budget on the performance of SPPE and SPLE. By setting the parameter $m$ to either $10^4$ or $10^5$, we can adjust the privacy loss budget to $\epsilon = 1$ or $\epsilon = 0.1$, respectively. Figure 8 illustrates the posterior approximation accuracy under different $\epsilon$ values. SPPE demonstrates rapid and stable convergence, while SPLE requires approximately 5 training rounds to converge.

### E.1.2 Inference on real disease outbreaks

We applied our inference methods (SPPE and SPLE) to analyze several real infectious disease outbreaks, namely influenza, Ebola, and COVID-19. For these experiments, we chose the privacy parameters $M = 100$, $N = 100$, and $L = 10$, which result in an overall privacy loss budget $\epsilon = 10$ according to Proposition 4.

**influenza outbreak.** We utilized the dataset from a boarding school, obtained from https://search.r-project.org/CRAN/refmans/epimdr/html/flu.html. The total population in the school was $K = 763$.

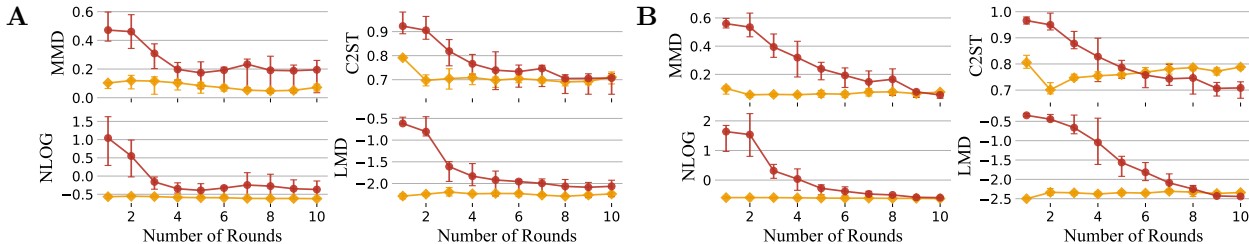

Figure 8: Approximation accuracy by SPPE (orange) and SPLE (red) against the number of rounds with **A.** Privacy loss budget $\epsilon = 1$. **B.** Privacy loss budget $\epsilon = 0.1$. The error bars represent the mean with the upper and lower quartiles.

We considered a daily time interval, and the observed private statistic $s_{\mathrm{dp}}^o$ is

$$s_{\mathrm{dp,flu}}^o = (0.0010,\ 0.0039,\ 0.0105,\ 0.0367,\ 0.0996,\ 0.2910,\ 0.3840,\ 0.3368,\ 0.3106,\ 0.2516).$$

**Ebola outbreak in West Africa, 2014.** We analyzed datasets from three regions: a) Guinea, b) Liberia, and c) Sierra Leone. The dataset source is from `https://apps.who.int/gho/data/node.ebola-sitrep`. We assumed potential contact individuals of $K = 100,000$. We selected 9 equally spaced time intervals of 120 days starting from 03/31/2014. The resulting observed private statistic $s_{\mathrm{dp}}^o$ is as follows

$$s_{\mathrm{dp,(a)}}^o = (0.0010,\ 0.0111,\ 0.0085,\ 0.0129,\ 0.0510,\ 0.0520,\ 0.0224,\ 0.0212,\ 0.0084,\ 0.0023).$$
$$s_{\mathrm{dp,(b)}}^o = (0.0010,\ 0.0007,\ 0.0019,\ 0.0664,\ 0.2579,\ 0.0742,\ 0.0610,\ 0.0542,\ 0.0172,\ 0.0003).$$
$$s_{\mathrm{dp,(c)}}^o = (0.0010,\ 0.0079,\ 0.0434,\ 0.2156,\ 0.2054,\ 0.0928,\ 0.0549,\ 0.0366,\ 0.0237,\ 0.0232).$$

**COVID-19.** We examined the COVID-19 dataset for Clark County, Nevada. See `https://usafacts.org/visualizations/coronavirus-covid-19-spread-map/state/nevada/county/clark-county/`. We assumed a potential contact population of $K = 100,000$. We selected 9 equally spaced time intervals of 24 days, starting from 09/07/2020. To obtain the number of currently infected individuals, we calculated the difference in the total confirmed cases with a time interval of 14 days from the original dataset. The resulting observed private statistic $s_{\mathrm{dp}}^o$ is:

$$s_{\mathrm{dp,covid}}^o = (0.0010,\ 0.0281,\ 0.0566,\ 0.0978,\ 0.2108,\ 0.2443,\ 0.2574,\ 0.0864,\ 0.0434,\ 0.0263).$$

The numerical results are presented and compared in Figure 4 of the main text.

### E.2 Bayesian linear regression

**Data generating parameters.** Following the parameters settings in Ju et al. (2022), we model the predictors with $x_{0,i} \sim \mathcal{N}_p(m, \Sigma)$, where $\Sigma = I_n$ and $m = (0.9,\ -1.17)$. The outcomes $y$ satisfy $y \mid x_0 \sim \mathcal{N}_n(x\beta, \sigma^2 I_n)$ where $\sigma^2 = 2$, and the prior for $\beta$ is independent normal $\mathcal{N}(0, 1)$. We set the privacy loss budget $\epsilon = 10$ and the number of subjects $n = 100$. The ground truth parameters are denoted as

$$\theta^* = (-1.79,\ -2.89,\ -0.66).$$

We simulate the summary statistics $\tilde{s}$ from the model using the ground truth parameters $\theta^*$, resulting in

$$\tilde{s} = \left( \begin{pmatrix} -0.3742 \\ -0.0629 \\ 0.0299 \end{pmatrix},\ 0.2499,\ \begin{pmatrix} 1.0000 & 0.0938 & -0.1270 \\ 0.0938 & 0.0180 & -0.0094 \\ -0.1270 & -0.0094 & 0.0280 \end{pmatrix} \right),$$

its corresponding vector form is

$$\tilde{s}_{\mathrm{vec}} = (-0.3742,\ -0.0629,\ 0.0299,\ 0.2499,\ 0.0938,\ -0.1270,\ 0.0180,\ -0.0094,\ 0.0280).$$

Furthermore, the observed private statistic $s_{\mathrm{dp}}^o$ is given by

$$s_{\mathrm{dp}}^o = \left( \left( \begin{array}{c} -0.3824 \\ -0.0667 \\ 0.0320 \end{array} \right), \ 0.2720, \ \left( \begin{array}{ccc} 1.0000 & 0.0988 & -0.1385 \\ 0.0988 & 0.0219 & -0.0229 \\ -0.1385 & -0.0229 & 0.0341 \end{array} \right) \right),$$

its corresponding vector form is

$$s_{\mathrm{dp,vec}}^o = (-0.3824, \ -0.0667, \ 0.0320, \ 0.2720, \ 0.0988, \ -0.1385, \ 0.0219, \ -0.0229, \ 0.0341).$$

**Experimental results.** In Figure 9, we present a comparison of the performance of the SPPE and SPLE methods across different metrics as the number of simulation rounds increases. The SPLE method demonstrates a faster convergence in this task. Figure 10 displays the posterior after 10 rounds, where both the SPPE and SPLE methods achieve results that are close to the near exact posterior obtained by the SMC-ABC method.

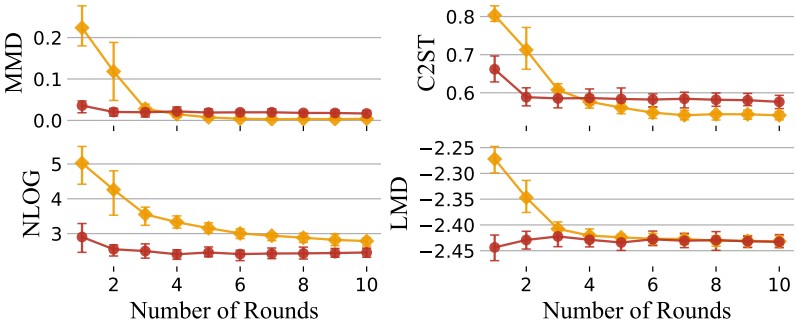

Figure 9: Approximation accuracy by SPPE (orange) and SPLE (red) on the Bayesian linear regression model against number of rounds, the error bars represent the mean with the upper and lower quartiles.

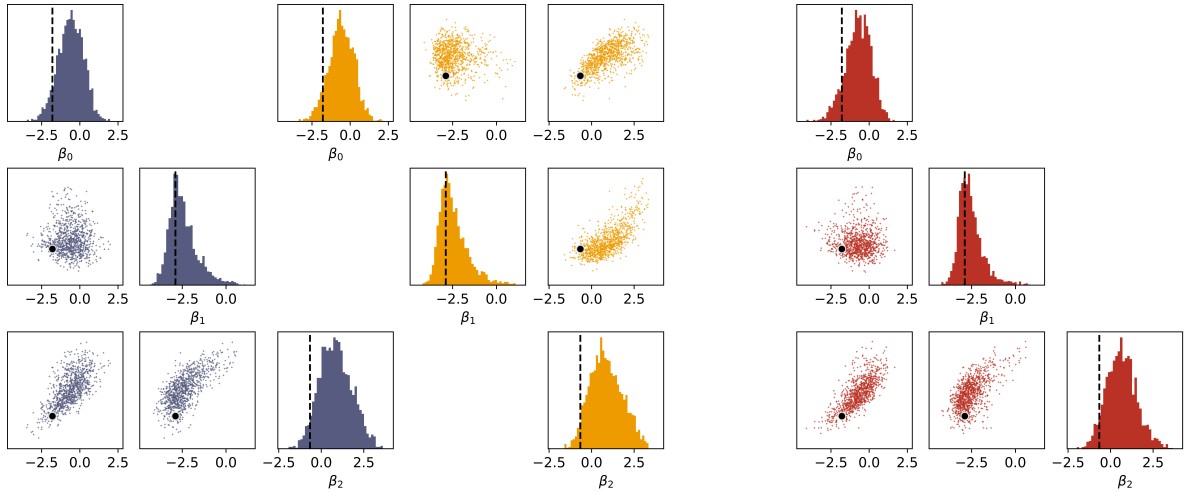

Figure 10: Posterior comparison on the Bayesian linear regression model. Grey: SMC-ABC; orange: SPPE; red: SPLE. The vertical lines and black dots indicate true data generating parameters.

To further characterize the privacy-utility trade-off, we compare the private data posterior distributions under several levels of privacy loss budget, in Figure 11. The underlying confidential data $x$ is the same for each $\epsilon = 0.1, 0.3, 1, 3, 10, 30, 100$. For each privacy loss level, we simulate one private summary $s_{\mathrm{dp}}(\epsilon)$, and use SPPE and SPLE to approximate the private data posterior $\pi(\theta \mid s_{\mathrm{dp}}; \epsilon)$. The two methods yield very

similar results. More interestingly, we use the same confidential data $x$ as Ju et al. (2022) and the posterior distributions in Figure 11 follow a similar trend with that in Figure 2 of Ju et al. (2022). For larger privacy loss budget (smaller noise), confidential data are mainly corrupted by clamping, and the proposed methods can elevate the effect of this censoring bias, as $\mathbb{E}(\theta \mid s_{\mathrm{dp}})$ is closer to the confidential data expectation $\mathbb{E}(\theta \mid x)$. As $\epsilon$ gets closer to 0 (near perfect privacy), the privacy mechanism has injected so much noise into $s_{\mathrm{dp}}$ that the posterior distribution is more dispersed, and little information about $\theta$ can be learned based on observing $s_{\mathrm{dp}}$.

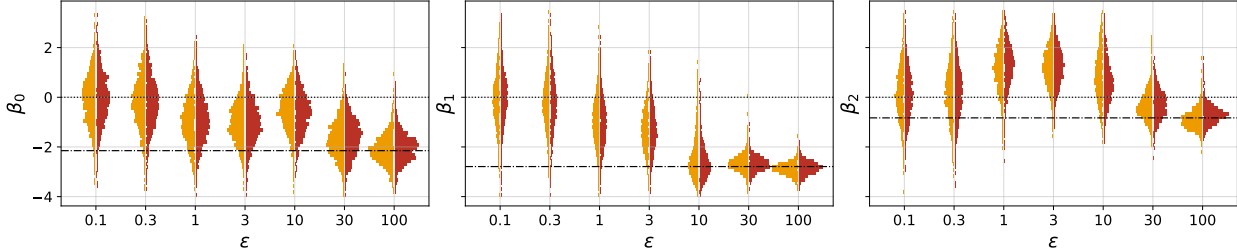

Figure 11: Marginal posterior histograms of $\theta = (\beta_0, \beta_1, \beta_2)$ given $s_{\mathrm{dp}}$ generated with several levels of privacy loss budget on the same confidential data $x$. Orange: SPPE; red: SPLE. The dash-dotted horizontal lines indicate the confidential data posterior means, and the dotted lines indicate prior means.

Finally, we compare the posterior means and 95% credible intervals obtained by different methods under various privacy loss budgets, as shown in Tables 2, 3, and 4. When larger noise is added, the posterior from all methods converge toward the prior of the parameters. Additionally, the naive posterior approximation fails to compute results due to the non-positive definiteness of the covariance matrix. The corresponding Kolmogorov-Smirnov test results, presented in Figure 12, shows that RNPE exhibits a larger bias compared to other methods, while the results from DA-MCMC, SMC-ABC, SPPE, SPLE, and Gibbs-SS are closely aligned.

Table 2: Estimated posterior mean and 95% credible intervals for the linear regression example using various methods. Here privacy loss budget is set to $\epsilon = 10$.

|  | $\beta_0$ | $\beta_1$ | $\beta_2$ |
|---|---|---|---|
| Confidential Posterior given $x$ | -2.15 (-2.68, -1.61) | -2.79 (-3.08, -2.50) | -0.83 (-1.08, -0.58) |
| Naive posterior approximation | -4.63 (-5.04, -4.22) | -6.23 (-6.57, -5.90) | -5.10 (-5.40, -4.79) |
| DA-MCMC | -0.62 (-2.50, 0.99) | -2.72 (-3.74, -0.96) | 0.54 (-1.06, 2.46) |
| SMC-ABC | -0.59 (-2.28, 0.93) | -2.44 (-3.67, -0.38) | 0.85 (-0.88, 2.77) |
| SPPE | -0.51 (-2.25, 1.07) | -2.40 (-3.61, -0.30) | 0.90 (-0.89, 2.85) |
| SPLE | -0.64 (-2.31, 0.96) | -2.61 (-3.65, -0.98) | 0.64 (-0.93, 2.48) |
| Gibbs-SS | -0.46 (-2.22, 1.39) | -0.50 (-2.08, 1.28) | 0.41 (-1.68, 2.13) |
| RNPE | -1.40 (-3.59, 0.94) | -2.15 (-3.34, 0.51) | 0.29 (-2.11, 2.97) |

Table 3: Estimated posterior mean and 95% credible intervals for the linear regression example using various methods. Here privacy loss budget is set to $\epsilon = 1$.

|  | $\beta_0$ | $\beta_1$ | $\beta_2$ |
|---|---|---|---|
| Confidential Posterior given $x$ | -2.15 (-2.68, -1.61) | -2.79 (-3.08, -2.50) | -0.83 (-1.08, -0.58) |
| Naive posterior approximation | - | - | - |
| DA-MCMC | -1.00 (-2.75, 0.85) | -0.96 (-2.66, 0.89) | 1.23 (-0.76, 2.88) |
| SMC-ABC | -0.80 (-2.74, 1.17) | -0.83 (-2.64, 1.23) | 0.90 (-1.00, 2.58) |
| SPPE | -0.93 (-2.80, 1.04) | -0.90 (-2.62, 0.97) | 1.11 (-0.71, 2.72) |
| SPLE | -0.89 (-2.69, 1.06) | -1.04 (-2.82, 0.91) | 1.25 (-0.55, 2.88) |
| Gibbs-SS | -0.92 (-2.72, 0.94) | -0.67 (-2.21, 1.04) | 1.04 (-0.62, 2.47) |
| RNPE | -1.37 (-3.81, 2.07) | -0.51 (-3.01, 2.53) | 1.34 (-1.77, 3.76) |

Table 4: Estimated posterior mean and 95% credible intervals for the linear regression example using various methods. Here privacy loss budget is set to $\epsilon = 0.1$.

|  | $\beta_0$ | $\beta_1$ | $\beta_2$ |
|---|---|---|---|
| Confidential Posterior given $x$ | -2.15 (-2.68, -1.61) | -2.79 (-3.08, -2.50) | -0.83 (-1.08, -0.58) |
| Naive posterior approximation | - | - | - |
| DA-MCMC | -0.12 (-1.98, 1.77) | -0.13 (-2.13, 1.88) | 0.13 (-1.91, 2.12) |
| SMC-ABC | -0.09 (-1.97, 1.93) | -0.10 (-2.09, 1.95) | 0.11 (-1.90, 2.22) |
| SPPE | -0.12 (-2.13, 1.73) | -0.12 (-2.11, 1.83) | 0.14 (-1.81, 2.09) |
| SPLE | -0.04 (-2.01, 1.99) | -0.06 (-2.00, 1.85) | 0.12 (-1.86, 2.09) |
| Gibbs-SS | -0.05 (-1.94, 1.95) | -0.01 (-1.95, 1.97) | 0.21 (-1.62, 1.89) |
| RNPE | -0.22 (-3.56, 3.16) | 0.13 (-3.12, 3.51) | 0.34 (-3.12, 3.61) |

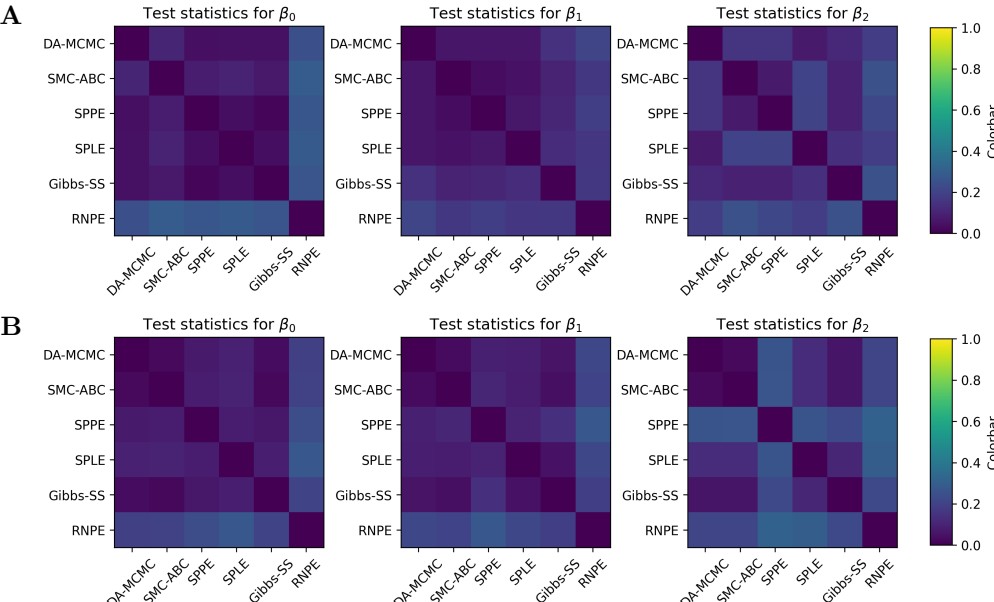

Figure 12: Kolmogorov-Smirnov test statistics between approximations of posterior marginals. **A.** Privacy loss budget $\epsilon = 1$. **B.** Privacy loss budget $\epsilon = 0.1$.

### E.3 Naïve Bayes log-linear model

**Model description.** The naïve Bayes log-linear model is a commonly used approach for modeling categorical data (Karwa et al., 2015). The input feature-vector, denoted as $x = (x_1, \cdots, x_K)$, consists of $K$ features, each taking values in the range $\{1, 2, \cdots, J_k\}$. The output class, denoted as $y$, represents the target category and takes values in $\{1, 2, \cdots, I\}$. The model assumes that the conditional probability of the input given the output, denoted as $p(x \mid y)$, can be factorized as the product of individual feature probabilities: $p(x \mid y) = \prod_{k=1}^{K} p(x_k \mid y)$. The model parameters are $p_{ij}^k$, which represents the probability $p(x_k = j \mid y = i)$, with prior $(p_{i,1}^k, \cdots, p_{i,J_k}^k) \sim \text{Dirichlet}(\alpha_{i,1}^k, \cdots, \alpha_{i,J_k}^k)$ for all $i$ and $k$; and $p_i = p(y = i)$, with prior $(p_1, \cdots, p_I) \sim \text{Dirichlet}(\alpha_1, \cdots, \alpha_I)$.

We assume that $(n_{i,1}^k, \cdots, n_{i,J_k}^k) \sim \text{Multinomial}(n_i; p_{i,1}^k, \cdots, p_{i,J_k}^k)$ for all $i$ and $k$ and $(n_1, \cdots, n_I) \sim \text{Multinomial}(n; p_1, \cdots, p_I)$. Here $n_{i,j}^k$ represents the counts $\#(y = i, x_k = j)$. One sufficient statistics of the model is the proportion of the counts $r_{i,j}^k := \frac{1}{n} n_{i,j}^k$, where $n = \sum_{i=1}^{I} \sum_{j=1}^{J_k} n_{i,j}^k$ for all $k$. To protect the privacy of the dataset, Laplace noise $e_{i,j}^k$ is added to the proportion of the counts, resulting in the privatized proportion $m_{i,j}^k = r_{i,j}^k + e_{i,j}^k$. When $e_{i,j}^k \sim \text{Laplace}(0, \frac{2K}{n\epsilon})$, the released private statistic $\{m_{i,j}^k\}_{i,j,k}$ satisfied $\epsilon$-DP.

In our simulation, we set $\alpha_{i,j}^k = \alpha_i = 2$ for all $i, j, k$, and $n = 100$, with $I = 2$, $K = 2$ and $J_k = 2$ for all $k$. The privacy loss budget $\epsilon = 10$. The ground truth parameters are

$$p_{1,1}^1 = 0.3887, \; p_{1,2}^1 = 0.6113, \; p_{1,1}^2 = 0.7537, \; p_{1,2}^2 = 0.2463,$$
$$p_{2,1}^1 = 0.6534, \; p_{2,2}^1 = 0.3466, \; p_{2,1}^2 = 0.5834, \; p_{2,2}^2 = 0.4166,$$
$$p_1 = 0.8489, \; p_2 = 0.1511.$$

and the observed private statistics simulated from the model with ground truth parameters are

$$r_{1,1}^1 = 0.3275, \; r_{1,2}^1 = 0.4520, \; r_{1,1}^2 = 0.5862, \; r_{1,2}^2 = 0.1827,$$
$$r_{2,1}^1 = 0.1293, \; r_{2,2}^1 = 0.0858, \; r_{2,1}^2 = 0.1288, \; r_{2,2}^2 = 0.0954.$$

**Experimental results.** Figure 13 illustrates the performance of the SPPE and SPLE methods across four different metrics, and Figure 14 shows the marginal posterior histograms after 10 rounds, our methods stabilize in performance after round 3.

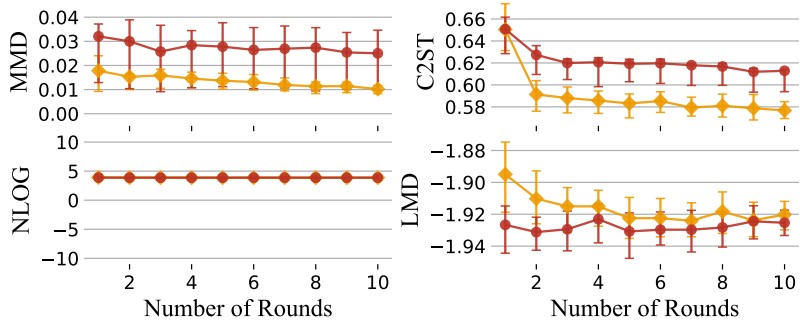

Figure 13: Approximation accuracy by SPPE (orange) and SPLE (red) on the log-linear model against the number of rounds, the error bars represent the mean with the upper and lower quartiles.

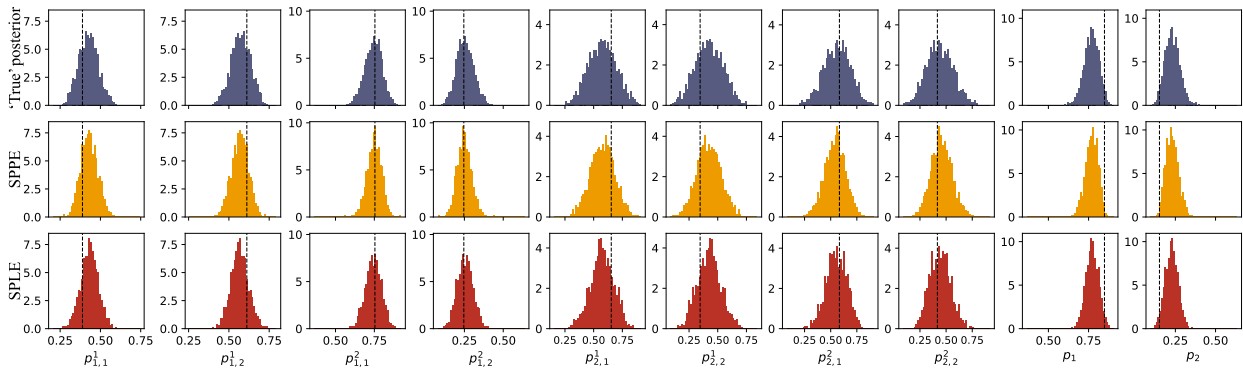

Figure 14: Marginal posterior histograms of the log-linear model. Grey: SMC-ABC; orange: SPPE; red: SPLE. The vertical lines indicate true data generating parameters.

### E.4 Neural spline flows with different layers

To evaluate the impact of varying the number of layers in the neural spline flow (NSF) model, we conducted additional experiments using configurations with 5 and 10 layers, in addition to the existing 8-layer setup.

The results are shown in Figure 15. For the Bayesian linear regression task, the posterior approximation performance with 5-layer and 10-layer configurations is comparable to the 8-layer NSF. This indicates that

a 5-layer NSF possesses sufficient flexibility to represent the posterior or likelihood. However, for the SIR model, the SPLE method with the 5-layer configuration converges slightly slower compared to the 8-layer NSF, indicating that fewer layers may marginally affect convergence in some cases. Besides, as analyzed in Appendix D, increasing the number of layers also results in a linear increase in computational cost.

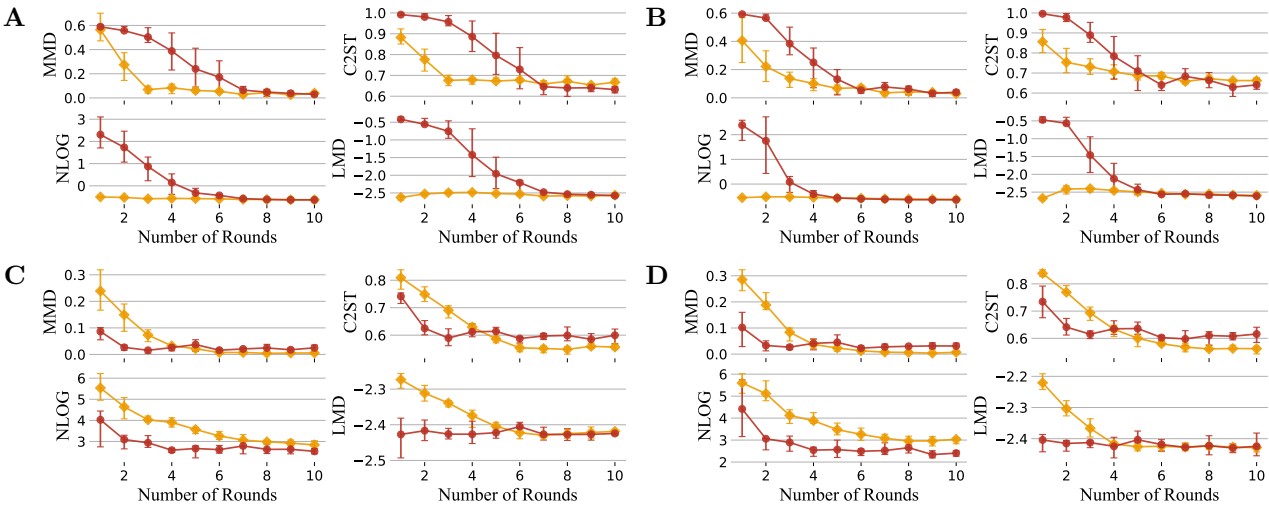

Figure 15: Comparison of approximation performance for NSF models with different layer configurations by SPPE (orange) and SPLE (red). **A.** 5-layer NSF on SIR model. **B.** 10-layer NSF on SIR model. **C.** 5-layer NSF on Bayesian linear regression model. **D.** 10-layer NSF on Bayesian linear regression model.

## F  Statement on Computing Resources

Our numerical experiments were conducted on a computer equipped with four GeForce RTX 2080 Ti graphics cards and a pair of 14-core Intel E5-2690 v4 CPUs.

