# OpenReview forum: "Simulation-based Bayesian Inference from Privacy Protected Data"
_TMLR — Accepted by TMLR_

### Review · Reviewer_Yzv4 · 2024-11-14

**Summary Of Contributions:**

This paper is concerned with statistical analysis of the data that has been privatized with privacy-protected data e.g. those obtained from differential privacy (DP) techniques. The authors' method assumes that the DP technique must be known and gives rise to the conditional density $\eta(s_{\text{dp}} | x)$ where $x$ is the private data and $s_{\text{dp}}$ is the DP output. As with typical Bayesian inference, the goal is to evaluate the posterior likelihood $\pi(\theta | s_{\text{dp}}) \propto f(s_{\text{dp}} | \theta) \pi(\theta)$ where $f(s_{\text{dp}} | \theta) = \int f(x | \theta) \eta(s_{\text{dp}} | x) \ dx$. The obstacle here is the intractability of $\pi(\theta | s_{\text{dp}}), f(s_{\text{dp}} | \theta)$ and possibly $f(x | \theta)$. To this end, the authors propose to approximate the posterior using sequential Monte Carlo approximate Bayesian computation (SMC-ABC), Sequential private-data posterior estimation (SPPE) and Sequential private-data likelihood estimation (SPLE).

**Audience:**

Yes

**Broader Impact Concerns:**

There is no ethical implications as far as I am concerned.

**Claims And Evidence:**

No

**Requested Changes:**

Critical changes:
- I would recommend this paper if the authors could address all of my concerns in the weaknesses, as doing so would increase the readability and the credibility of the paper.
- Note that $\pi(\theta | s) = \pi(\theta | x)$ when $s$ is a sufficient statistics, and that's why we are interested in calculating $\pi(\theta | s)$. The authors should add a brief discussion about this point.
- Maybe I couldn't find it, but what is the value of $\epsilon$ in the first experiment?

Minor changes:
- Can the authors add some explanation on why they choose the neural spline flows for the approximations? I am curious if there are any other models that the authors have tried but resulted in poor approximation quality.

**Strengths And Weaknesses:**

Strengths:
- I believe that the results from this paper could benefit privacy researchers: Assuming that the data have already been privatized by a DP mechanism, the proposed method presents us a logical way to approximate the Bayesian posterior.
- Demonstrations of the use of these methods for real world applications are appreciated.
- The experimental results indicate that the proposed methods (SPPE and SPLE) can accurately approximate the baseline (AMC-ABC), though SPPE seems to perform better overall.

Weaknesses:
- From what I understand, the Bayesian inference is the main theme for this paper, but the paper's title does not reflect this. At least, Bayesian inference should has its own subsection in the background section. Authors could argue that this work applies in non-Bayesian setting as well, in which case I recommend the authors to provide at least one example of such setting.
- The presentation in Section 3 (Methods) are confusing: In the first paragraph, the authors wrote "We present two complementary
approaches..." without telling the readers which subsections refer to these two approaches---this leaves the readers to do some pattern matching of the two approaches to the three subsections (excluding the algorithm section).
- The SMC-ABC method is the main method, but is barely talked about in this paper. In some places, the method is referred to as the "baseline"; in the experiment, it is referred to as "'True' posterior". (I was quite confused about this term until I read the experiment details in the Appendix "..we performed simulations up to 5 × 10^5 times for the SMC-ABC method to generate the near exact ‘True’ posterior"...). If SMC-ABC is one of the main methods proposed to solve the problem, the authors should run its algorithm with the same simulation steps as those of SPPE and SPLE and show the results in the main paper.
- The discussions of the experiments results are quite short. Specifically, why is SPPE generally better than SPLE? Under what condition does SPLE perform relatively well? Does the number/width of layers affect the approximation quality?

---

> ### Author Response · Authors · 2025-02-07
> **Response**
>
> Dear reviewer Yzv4, we sincerely thank you for your detailed advice and the time you spent on our paper. Here are our replies to your comment:
>
> ---
>
> **Weakness:**
>
> **W1:** Bayesian inference is the main theme for this paper, but the paper's title does not reflect this.
>
> **A1:** Thank you for raising this issue. We agree that Bayesian inference forms the backbone of our work. To better reflect this focus, we have revised the title to "Simulation-based Bayesian Inference from Privacy Protected Data", and added an overview of Bayesian inference in the background section.
>
> **W2:** The authors wrote "We present two complementary approaches..." without telling the readers which subsections refer to these two approaches.
>
> **A2:** Thank you for pointing this out. We have revised the manuscript to make the connections between the approaches and the subsections clearer to avoid confusion.
>
> > We present two complementary approaches: a) Section 3.2: approximate the private data marginal likelihood $f(s_{\mathrm{dp}} \mid \theta)$ in equation 2, and b) Section 3.3: approximate $\pi(\theta \mid s_{\mathrm{dp}})$ from equation 3 directly.
>
> **W3:** The SMC-ABC method is barely talked about in this paper. In some places, the method is referred to as the "baseline"; in the experiment, it is referred to as "True posterior".
>
> **A3:** Thank you for raising this issue. We have revised the background section (2.1), method section (3.1), and description of the SMC-ABC algorithm (Appendix C) to include a more detailed discussion and deployment of the SMC-ABC method.
>
> Besides, we have clarified why SMC-ABC is referred to as both the "baseline" and the "True posterior" in the revised manuscript. When used with a limited number of simulations (e.g., $10^4$), SMC-ABC serves as a baseline for performance comparison, though its approximation may not be ideal due to using fewer samples. However, with a significantly larger number of simulations (e.g., $5\times 10^5$), SMC-ABC can achieve near-exact posterior results. In these cases, we use it as a reference for the "True posterior".
>
> **W4:** Why is SPPE generally better than SPLE? Under what condition does SPLE perform relatively well?
>
> **A4:** The performance of SPPE and SPLE depends on the specific task. In our experiments, SPPE performed better in the SIR model and log-linear model, while SPLE performed better in Bayesian linear regression. We think this difference depends on the likelihood-free model structure, such as the dimensionality and complexity of the model posterior or likelihood. As discussed in [2], there is no clear-cut answer as to which method (posterior estimations or likelihood estimations) should be preferred.
>
> **W5:** Does the number/width of layers affect the approximation quality?
>
> **A5:** Thank you for raising this point. We have added results in Appendix E.4 to include comparisons across different numbers of layers (5 and 10 layers). The results are comparable to the current setting (8 layers).
>
> We also highlight that our proposed methods are not tied to a specific implementation of the flow structure. Instead, the novelty lies in the proposed loss functions for differential privacy settings (Eq.(5) and Eq.(8)), which are independent of the specific flow structure used.
>
> ---
>
> **Requested Changes:**
>
> **R1:** The authors should add a brief discussion about sufficient statistics.
>
> **A1:** Thank you for your suggestion. We have added a discussion about sufficient statistics in the background of the revised manuscript.
>
> > We exploit output perturbation on summary statistics in two ways.
> > From an inference perspective, when $s(x)$ is a sufficient statistic for $\theta$, the confidential data posterior $\pi(\theta \mid x)$ is fully characterized by $s(x)$ and the prior. This is the rationale for choosing $\pi(\theta \mid s_\mathrm{dp})$ as the target distribution and for the posterior from perturbed data to be still informative about the model parameters. From a computation perspective, since many summary statistics are of lower dimension than the confidential data, this can facilitate efficient computations. In Section 3.4, we leverage the low dimensionality of $s_\mathrm{dp}$ by incorporating quasi-Monte Carlo techniques.
>
> **R2:** What is the value of $\epsilon$ in the first experiment?
>
> **A2:** We thank the reviewer for this important suggestion. In the first experiment, the differential privacy parameter $\epsilon$ is set to 10, as stated in Appendix E.1.1. We now explicitly state the DP parameters in the first experiment of the main text.
>
> > We set $L=10$ time points with $n=1000$ and $m=1000$, achieving $\epsilon=10$ differential privacy. Lower privacy loss budgets ($\epsilon=1,0.1$) are explored in Appendix E.1.1 by adjusting $m$ accordingly.

---

> ### Author Response · Authors · 2025-02-07
> **Response Contd.**
>
> **R3:** Why authors choose the neural spline flows for the approximations? I am curious if there are any other models that the authors have tried but resulted in poor approximation quality.
>
> **A3:** We chose neural spline flows (NSFs) because they are one of the most widely used and flexible neural conditional density structures for density approximation tasks [1, 2]. NSFs offer higher flexibility compared to other models and it can capturing complex distributions [3].
>
> In our earlier experiments, we also tested masked autoregressive flows (MAFs) [4], which have fewer parameters but are less flexible. MAFs performed worse than NSFs in terms of the C2ST metric. While we did not include MAF results in the final manuscript to maintain focus on the best-performing models, we have provided the implementation of MAFs in our code repository for reference.
>
> ---
>
> [1]. Kelly et al., "Misspecification-robust Sequential Neural Likelihood", 2023.
>
> [2]. Lueckmann et al., "Benchmarking Simulation-Based Inference", 2021.
>
> [3]. Durkan et al., "Neural Spline Flows", 2019.
>
> [4]. Papamakarios et al, "Masked Autoregressive Flow for Density Estimation", 2017.

---

> > ### Comment · Reviewer_Yzv4 · 2025-03-09
> > **Thank you**
> >
> > Thank you for taking the effort to answer my concerns. I am satisfied by the answers. This looks like a really nice paper.

---

### Review · Reviewer_u4vF · 2024-12-16

**Summary Of Contributions:**

The authors propose three simulation-based methods for likelihood-free inference to estimate population parameters from data protected by privacy measures. These include Sequential Monte Carlo for Approximate Bayesian Computation, as well as two techniques based on neural density estimation: Sequential Private-Data Posterior Estimation and Sequential Private-Data Likelihood Estimation. They validate these methods with numerical experiments, applying them to the Susceptible-Infected-Recovered (SIR) model—augmented with a privacy mechanism—and to Bayesian linear regression.

**Audience:**

Yes

**Claims And Evidence:**

Yes

**Requested Changes:**

- (Critical) Can you update the title to reflect the problem that is being studied ?
- Can you add mode details on the background on MCMC methods and normalizing flows ? The paper treats the two communities that it targets itself to (i.e. Differential Privacy and simulation-based statistics) differently, first by taking the time to detail many basic facts for people from differential privacy, but then by assuming a lot of implicit prior knowledge on the simulations parts.
- Is it possible to have the results for experiments with realistic $\epsilon$s ? Between $0.1$ and $1$ for instance.

**Strengths And Weaknesses:**

Strengths:

- The paper addresses a significant and seemingly novel problem, contributing to an important area of research.
- The introduction and privatization of the SIR model is a valuable and noteworthy addition.
- The study demonstrates strong empirical results, which lend credibility to the proposed methods.

Weaknesses:

- The article’s writing could be improved, as it is challenging to follow for readers unfamiliar with the presented concepts.
- The title does not effectively convey the core problem studied in the paper.
- The choice of $\epsilon = 10$ in the simulations is questionable, as this value is unusually high.
- The scalability of the proposed method with increasing dimensionality has not been thoroughly explored by the authors.

---

> ### Author Response · Authors · 2025-02-07
> **Response**
>
> Dear reviewer u4vF, we sincerely thank you for your detailed advice and the time you spent on our paper. Here are our replies to your comment:
>
> ---
>
> **Weakness:**
>
> **W1:** The title does not effectively convey the core problem studied in the paper.
>
> **A1:** Thank you for the suggestion. As you noted, our paper focuses on likelihood-free inference to estimate population parameters from data protected by privacy measures. To better reflect this theme, we have updated the title to "Simulation-based Bayesian Inference from Privacy Protected Data".
>
> **W2:** The choice of $\epsilon=10$ in the simulations is questionable, as this value is unusually high.
>
> **A2:** Thank you for pointing this out. In response, we have added results for $\epsilon=1$ and $\epsilon=0.1$ in Appendix D. Under these settings for SIR model, SPPE still shows rapid and stable convergence, while SPLE requires approximately 5 training rounds to converge. As for Bayesian linear regression, we compared the estimated posterior means, 95% credible intervals, and Kolmogorov-Smirnov test statistics under different $\epsilon$ values. Our methods are still close to the reference posterior under small privacy budgets.
>
> **W3:** The scalability of the proposed method with increasing dimensionality has not been thoroughly explored.
>
> **A3:** We acknowledge that increasing dimensionality presents a challenge for inference with privacy data. To address your concerns, we have added a detailed computational complexity analysis in Appendix D: optimizing the flow parameters has a complexity that scales linearly with the dimensionality of $\theta$ and $s$ and quadratically with the number of nodes in the neural network. Also, in many practical scenarios (e.g. the SIR and Bayesian linear regression models), the dimensionality of $s$ remains constant regardless of the confidential data sample size $n$. Since we have defined neural density estimators for $f(s_{\mathrm{dp}} \mid \theta)$ and $\pi(\theta \mid s_{\mathrm{dp}})$, the dimension of the input \& output layers of the normalizing flow is not affected by the sample size $n$.
>
> For cases where the dimensionality of the summary statistics $s$ or parameters $\theta$ is higher, we have included a 10-dimensional log-linear example in Appendix E.3 to demonstrate the applicability of our method to higher-dimensional tasks.
>
> ---
>
> **Requested Changes:**
>
> **R1:** Can you update the title to reflect the problem that is being studied?
>
> **A1:** We have updated the title to reflect the main theme. Please see **W1** above.
>
> **R2:** Can you add more details on the background on MCMC methods and normalizing flows?
>
> **A2:** Thank you for highlighting this. We have revised the background section to include details of SMC-ABC and neural density estimation methods for likelihood-free inference, along with the motivation for using normalizing flows for density estimation in Section 2.1. We also have introduced the likelihood-based MCMC method in Section 2.3.
>
> **R3:** Is it possible to have the results for $\epsilon$ between 0.1 and 1?
>
> **A3:** We have added results for the SIR model and Bayesian linear regression under $\epsilon=1$ and $\epsilon=0.1$ in Appendix E. Please see **W2** above.

---

### Review · Reviewer_76WG · 2025-01-29

**Summary Of Contributions:**

The paper focuses on learning noise-aware posterior distributions from differentially private query results, when the likelihood function is not readily available but a simulator is. The authors propose to use normalizing flows to approximate either the (intractable) marginal likelihood, or the noise-aware posterior directly. They experimentally show that their method can achieve similar performance as the baselines (sequential Monte Carlo ABC method for SIR model on synthetic and real epidemic data, existing noise-aware methods for linear regression with simple synthetic data), while saving on the computation cost (compared to SMC-ABC).

**Audience:**

Yes

**Broader Impact Concerns:**

No concerns

**Claims And Evidence:**

No

**Requested Changes:**

1. Clarify writing in general; preferably introduce all background material in the corresponding section, not under Methods (introducing eg SMC-ABC, normalizing flows in the Background would make this clearer).
2. Contributions: mention explicitly in what sense the proposed methods are claimed to provide efficiency and utility (and preferably add ref to corresponding Sec that backs up the claim).
3. Clarify how the proposed method scales to higher dimensional $s_{dp}$ DP summary statistics (eg, based on the discussion in Sec3.3 & Sec4.1, it is not clear if the focus on low-dimensional queries is a fundamental limitation of the approach or simply a convenience).
4. Clarify how the SMC-ABC baseline (or proposed method?) is done.
5. Experimental details: mention how the hyperparameters have been chosen for each method (grids, Bayesian optimization, something else), also report the DP parameters clearly for all experiments in the main paper (SIR models).
6. Should add some missing references to the discussion in Related work:
	* Williams & McSherry 2010: Probabilistic inference and DP
		* first proposed using Bayesian inference to handle DP noise (should mention, no need to compare experimentally)
	* Kulkarni et al. 2021: DP Bayesian Inference for GLMs
		* propose method for noise-aware GLMs (should mention, no need to compare experimentally)
	* Lee et al. 2022: DP Normalizing Flows for Synthetic Tabular Data Generation
		* also combine normalizing flows with DP (should mention, no need to compare experimentally)
7. p6 just before Sec 4: "...achieve good accuracy with fewer samples from the simulator." Fewer samples compared to what, what is the concrete baseline here?

**Strengths And Weaknesses:**

### Strengths

* The general problem is interesting and important.
* The proposed methods seem to work for some problems for which the currently existing noise-aware methods are not applicable (intractable likelihood).


### Weaknesses

* The writing in general could be improved; most seriously, it takes some effort to try and distinguish novel contributions from existing work, as much of the actual background material is interleaved with the contributions in the Methods section (eg importance sampling to reduce variance on p4, basics of RQMC on p5)
* It is not entirely clear what some of the claimed contributions mean (see Requested changes for specifics).
* There is no mention of releasing the source code.
* Various other issues (see Requested changes for details).

---

> ### Author Response · Authors · 2025-02-07
> **Response**
>
> Dear reviewer 76WG, we sincerely thank you for your detailed advice and the time you spent on our paper. Here are our replies to your comment:
>
> ---
>
> **Weakness:**
>
> **W1:** The writing in general could be improved; Much of the actual background material is interleaved with the contributions in the Methods section.
>
> **A1:** Thank you for pointing this out. We have restructured the manuscript to better separate the background material from our methodological contributions. Specifically, we have expanded the Background section to include clear descriptions of SMC-ABC methodology and normalizing flows. Please refer to **R1** and **R2** below for details.
>
> Additionally, regarding the reference to "importance sampling to reduce variance on p4", we now explicitly state:
>
> > Based on this idea, we design a modified loss of equation 6 that leverages the APT framework to improve efficiency in posterior estimation. Specifically, we propose the following loss function.
>
> **W2:** There is no mention of releasing the source code.
>
> **A2:** Thank you for your reminder. We have now included the source code in the attachment for reference.
>
> ---
>
> **Requested Changes:**
>
> **R1:** Clarify writing in general; preferably introduce all background material in the corresponding section. Introducing SMC-ABC, normalizing flows in the Background would make this clearer.
>
> **A1:** Thank you for your suggestion.
> We have reorganized Section 2. It now begins with an introduction to likelihood-free inference methods (e.g. SMC-ABC and neural density estimations) in the confidential data setting, followed by a review of differential privacy. We conclude the section by explaining why inference based on privatized data is challenging.
>
> **R2:** Contributions: mention explicitly in what sense the proposed methods are claimed to provide efficiency and utility.
>
> **A2:** Thank you for your suggestion. We have revised the Contributions to emphasize the efficiency of our methods. Specifically, our approach reduces the number of required simulations while maintaining inference accuracy. The updated section now states:
>
> > We demonstrate the efficiency of our methods on an infectious disease model using synthetic and several real disease outbreak data. SPPE and SPLE require fewer numbers of simulations from the data model to achieve the same inference results compared with SMC-ABC.
>
> **R3:** Clarify how the proposed method scales to higher dimensional. It is not clear if the focus on low-dimensional queries is a fundamental limitation of the approach or simply a convenience.
>
> **A3:** We have added a detailed computational complexity analysis in Appendix D: optimizing the flow parameters has a complexity that scales linearly with the dimensionality of $\theta$ and $s$ and quadratically with the number of nodes in the neural network. Also, in many practical scenarios (e.g. the SIR and Bayesian linear regression models), the dimensionality of $s$ remains constant regardless of the confidential data sample size $n$. Since we have defined neural density estimators for $f(s_{\mathrm{dp}} \mid \theta)$ and $\pi(\theta \mid s_{\mathrm{dp}})$, the dimension of the input \& output layers of the normalizing flow is not affected by the sample size $n$.
>
> For cases where the dimensionality of the summary statistics $s$ or parameters $\theta$ is higher, we have included a 10-dimensional log-linear example in Appendix E.3 to demonstrate the applicability of our method to higher-dimensional tasks.
>
> Besides, the choice of focusing on using summary statistics queries (and hence the lower dimension) is motivated by existing differential privacy mechanisms that add noise to summary statistics. In the revised Section 2.2, we describe how we "exploit output perturbation on summary statistics" from inference and computation perspectives. This low dimensionality is not a fundamental limitation of our approaches, which are designed to accommodate existing privacy mechanisms, many of which produce privatized low-dimensional summary statistics.
>
> **R4:** Clarify how the SMC-ABC baseline is done.
>
> **A4:** Thank you for your suggestion. In Section 2.1, we have added the background on the SMC-ABC method. To further clarify the implementation of the SMC-ABC baseline, we provide details on its adaptation to the private data setting in Section 3.1 and include a description of the algorithm in Appendix C.

---

> ### Author Response · Authors · 2025-02-07
> **Response Contd.**
>
> **R5:** Mention how the hyperparameters have been chosen for each method, also report the DP parameters clearly for all experiments in the main paper (SIR models).
>
> **A5:** We thank the reviewer for this important suggestion. We have explicitly stated the DP parameters in SIR model of the main text.
>
> > We set $L=10$ time points with $n=1000$ and $m=1000$, achieving $\epsilon=10$ differential privacy. Lower privacy loss budgets ($\epsilon=1,0.1$) are explored in Appendix E.1.1 by adjusting $m$ accordingly.
>
> We also modified the selection of hyperparameters in the flow structure in Appendix E.
>
> > Following the flow structure settings in (Lueckmann et al., 2021), each layer consists of two residual blocks with 50 units and ReLU activation function, with 10 bins in each monotonic piecewise rational-quadratic transforms, and the tail bound was set to 5.
> > Through empirical evaluation, we found that 8 layers provided sufficient flexibility for posterior estimation. Appendix E.4 compares 5-layer and 10-layer models, showing that 5 layers slightly degrade performance, while 10 layers perform similarly to 8 layers but with higher computational cost.
>
> **R6:** Should add some missing references to the discussion in Related works.
>
> **A6:** Thank you for your suggestion. We have updated the Related Works to provide a more comprehensive overview of relevant studies.
>
> **R7:** P6 just before Sec 4: "...achieve good accuracy with fewer samples from the simulator." Fewer samples compared to what, what is the concrete baseline here?
>
> **A7:** Thank you for your suggestion. The phrase "fewer samples from the simulator" refers to a comparison with methods that do not utilize sequential training. To clarify this, we have revised the manuscript accordingly, adding the clause:
>
> > compared with non-sequential training procedures that use a fixed proposal distribution.
>
> We also hope that the discussion on how SMC-ABC differs from ABC (Section 2.1) helps clarify that the comparison is between sequential and non-sequential training methods.
>
> ---
>
> [1]. Lueckmann et al., "Benchmarking Simulation-Based Inference", 2021.

---

> > ### Comment · Reviewer_76WG · 2025-02-18
> > **Some further discussion**
> >
> > Thank you for the update and the fixes, I think the paper is generally a lot clearer now. Here are some further comments and some questions:
> > * Final lines of p6: "If our neural approximation family ... has bounded Hardy-Krause variation...". I guess showing this analytically is not doable for the models you consider?
> > * Second paragraph p8: "Then the RQMC methods...can provide efficiency and accuracy...". This is related to the questions about efficiency and scalability I had in the original review: does this sentence mean that RQMC only works with low-dimensional $s_{dp}$, or how should I interpret this?
> > * I still do not easily notice what the DP level is in Fig 2 experiment. Now that I hopefully understand what the baseline here means, is it possible to also run a non-DP baseline for Fig 2?
> > * The cost of privacy, second paragraph, p10: "we highlight that it is necessary to design a valid inference procedure...". I though this is already a very well-understood point, as there are plenty of papers on the noise-aware inference? Writing this as though it is somehow novel observation seems odd. Same goes for the second point on the privacy-utility trade-off.
> > * Some things more suited for background are still in the main text (eg RQMC, which is as a drop-in replacement for standard MC as far as I can tell, is first introduced in Sec3.4). However, this now looks like a minor problem in the fixed version.
> >
> > ### Very minor comments
> >
> > No need to comment on these, simply fix when appropriate:
> > * Def 1 on p3: $eta \rightarrow \eta$
> > * just above Eq(2) on p4: $sdp \rightarrow s_{dp}$
> > * Above Eq(5) on p5 & start of Sec 3.3 on p5: let's $\rightarrow$ let us
> > * After Eq(10) on p6: "...Exponential \[mechanism\] can be easily simulated": the exponential mechanism in general is not easy to simulate?
> > * first paragraph, p19: "...loss on validation data does not increase..." $\rightarrow$ decrease?

---

> > > ### Author Response · Authors · 2025-02-21
> > > **Response**
> > >
> > > Thanks for your reply. We address your concerns and incorporate your suggestions below.
> > >
> > > **C1:** Final lines of p6: "If our neural approximation family ... has bounded Hardy-Krause variation...". I guess showing this analytically is not doable for the models you consider?
> > >
> > > **A1:** Thank you for raising this point. Verifying the bounded Hardy-Krause variation condition analytically depends on the specific neural density architecture. Recent work [1, Theorem 4.2] has shown that under certain assumptions, autoregressive flows (e.g., masked autoregressive flows [2]) can achieve an RMSE of order $\mathcal{O}(M^{-1+\epsilon})$ for arbitrarily small $\epsilon>0$, where $M$ is the RQMC sequence length. For neural spline flows, the analysis is more complex. Instead of a theoretical analysis, we chose to verify the convergence rate Eq. (12) empirically with Appendix E, Figure 7.
> > >
> > > **C2:** Does this sentence ["Then the RQMC methods...can provide efficiency and accuracy..."] mean that RQMC only works with low-dimensional $s_{dp}$, or how should I interpret this?
> > >
> > > **A2:** RQMC methods indeed perform well in low-dimensional spaces, but we cannot conclude about their performance in high dimensions. On one hand, RQMC performance may degrade when used in high-dimensional settings. On the other hand, RQMC may work for high-dimensional integrals when the integrand can be "well approximated by a sum of low-dimensional functions" [2, Section 3].
> > >
> > > The theoretical analysis in [3] shows that the discrepancy of RQMC sequences is of the order $\mathcal{O}(M^{-1}(\log M)^r)$,  where $M$ is the length of the sequence and $r$ is the dimension. As $r$ increases, the impact of the $(\log M)^r$ term becomes more significant, which may result in the root-mean-square error (RMSE) from achieving the ideal $\mathcal{O}(M^{-1})$ rate. To demonstrate the convergence rate in RMSE, we compared the RMSE order of the RQMC method on two tasks (see Appendix E, Figure 7). For the SIR model (with 10-dimensional summary statistics), the convergence rate of RQMC is approximately $\mathcal{O}(M^{-0.97})$, which is very close to the theoretical upper bound $\mathcal{O}(M^{-1})$. For the linear regression model (with 9-dimensional summary statistics), the convergence rate is about $\mathcal{O}(M^{-0.89})$. Although this is lower, it still outperforms the standard Monte Carlo method, which converges at $\mathcal{O}(M^{-0.5})$ (as also verified in Figure 7).
> > >
> > >
> > > **C3a:** I still do not easily notice what the DP level is in Fig 2 experiment.
> > >
> > > **C3b:** Is it possible to also run a non-DP baseline for Fig 2?
> > >
> > > **A3:** Thank you for your comment. The privacy level for Fig 2 is $\epsilon = 10$, which is the same as the synthetic data examples. We will add this information to the figure caption. We will also add a non-DP baseline to Fig 2. This non-DP baseline is obtained from the ordinary differential equations solution to the infection curve. However, the main purpose of the comparison is to compare results from SPLE and SPPE with the "true posterior" obtained from running long SMC-ABC chains.
> > >
> > > We state that "Our $R_0$ estimates using privatized data are consistent with common estimates of $R_0$ for these diseases" with a reference to [5]. We will revise this and explicitly state the range of these common $R_0$ estimates: $(1.51, 2.53)$ for Ebola, $(1.5, 3.5)$ for COVID-19. The exception is flu $(0.9, 2.1)$, which as mentioned "should be modeled by a Susceptible-Infected" model instead.
> > >
> > > **C4:** "The cost of privacy, p10", I though this is already a very well-understood point, as there are plenty of papers on the noise-aware inference. Writing this as though it is somehow novel observation seems odd. Same goes for the second point on the privacy-utility trade-off.
> > >
> > > **A4:** We did not intend to write this paragraph as new discoveries. We avoid any misunderstanding, we will begin it with "Our experiments demonstrate the privacy-utility trade-off, which has also been demonstrated by related works on noise/privacy-aware inference."

---

> > > ### Author Response · Authors · 2025-02-21
> > > **Response Contd.**
> > >
> > > **C5:** Some things more suited for background are still in the main text (eg RQMC, which is as a drop-in replacement for standard MC as far as I can tell, is first introduced in Sec3.4). However, this now looks like a minor problem in the fixed version.
> > >
> > > **A5:** We believe it is more natural to introduce RQMC in Section 3, after setting up our objective functions and the integration problem we use RQMC to solve. However, we will add a sentence "Later, in Section 3.4, we will use randomized quasi-Monte Carlo (RQMC) to improve standard Monte Carlo estimates", after we first introduce the idea of using Monte Carlo estimators to approximate integrals.
> > >
> > > **C6:** Some minor comments on typos.
> > >
> > > **A6:** Thank you for catching them. We have corrected the typos and removed the statement that "the Exponential mechanism can be easily simulated", as we agree that it is generally not so.
> > >
> > > ---
> > >
> > > [1]. Liu, S., "Transport quasi-Monte Carlo", 2024.
> > >
> > > [2]. Papamakarios, G., et al., "Masked autoregressive flow for density estimation", 2017.
> > >
> > > [3]. L’Ecuyer, P., "Randomized quasi-Monte Carlo: An introduction for practitioners", 2018.
> > >
> > > [4]. Owen, A., "Monte Carlo variance of scrambled net quadrature", 1997.
> > >
> > > [5]. Eisenberg. J., "R0: How Scientists Quantify the Intensity of an Outbreak Like Coronavirus and Its Pandemic Potential", 2020.

---

> > > > ### Comment · Reviewer_76WG · 2025-02-24
> > > > **No further questions**
> > > >
> > > > Thank you for the response, I have no further questions at this point. Please update the draft with all the discussed changes when you can.

---

> > > > > ### Author Response · Authors · 2025-03-10
> > > > > **Response**
> > > > >
> > > > > Thank you for your feedback. We have now updated the manuscript with all the discussed changes.

---

### Decision · Action_Editor_DbCy · 2025-03-24

**Recommendation:** Accept as is

**Comment:**

The authors have answered all reviewer concerns and all reviewers thus consider the paper to be acceptable to TMLR.

As a minor formatting point for the camera ready version, I would encourage the authors to refer to equations in the paper as "Equation (3)" instead of "equation 3", as is standard practice. Please also check the bibliography for consistency, inclusion of publication forums and correct capitalisation of terms derived from proper names (e.g. Bayesian, Gaussian) and abbreviations (e.g. GAN).

**Audience:**

All reviewers agree that the paper is interesting to the DP Bayesian inference community.

**Claims And Evidence:**

All reviewers agree that the submission is supported by sufficient evidence.

---

> ### Author Response · Authors · 2025-03-27
>
> Thank you very much for the support. We have uploaded the camera-ready version with the suggested edits.